# Unravelling and reconstructing the biosynthetic pathway of bergenin

Ruiqi Yan[1,5], Binghan Xie[1,5], Kebo Xie ✪[2,3,4,5] ✉, Qi Liu[1,5], Songyang Sui[2,3,4], Shuqi Wang[1], Dawei Chen[2,3,4], Jimei Liu[2,3,4], Ridao Chen[2,3,4], Jungui Dai ✪[2,3,4] ✉ & Lin Yang ✪[1] ✉

Bergenin, a rare *C*-glycoside of 4-*O*-methyl gallic acid with pharmacological properties of antitussive and expectorant, is widely used in clinics to treat chronic tracheitis in China. However, its low abundance in nature and structural specificity hampers the accessibility through traditional crop-based manufacturing or chemical synthesis. In the present work, we elucidate the biosynthetic pathway of bergenin in *Ardisia japonica* by identifying the highly regio- and/or stereoselective 2-*C*-glycosyltransferases and 4-*O*-methyltransferases. Then, in *Escherichia coli*, we reconstruct the de novo biosynthetic pathway of 4-*O*-methyl gallic acid 2-*C*-β-D-glycoside, which is the direct precursor of bergenin and is conveniently esterified into bergenin by in situ acid treatment. Moreover, further metabolic engineering improves the production of bergenin to 1.41 g L$^{-1}$ in a 3-L bioreactor. Our work provides a foundation for sustainable supply of bergenin and alleviates its resource shortage *via* a synthetic biology approach.

Bergenin (**1**), the main bioactive component of the traditional Chinese medicinal plant *Ardisia japonica*, has been widely used in the clinical treatment of chronic bronchitis and pulmonary tuberculosis[1,2]. Bergenin possesses diverse biological activities, including antitussive, antinociceptive, anti-inflammatory, anti-HIV, antidiabetic, and anticancer effects[3–6] indicating bergenin is an important natural product with multiple benefits to human health. To date, bergenin is mainly obtained by direct extraction from the rhizomes or roots of *Ardisia japonica, A. creanata, Bergenia crassifolia, B. purpurascens, Rodgersia sambucifolia*, etc.; however, this process is limited by the low and unstable content of bergenin in these medicinal plants[7–9]. Moreover, the chemical synthesis of bergenin faces many challenges, especially in terms of regio- and/or stereoselective *C*-sugar and *O*-methyl group introduction[10]. In recent years, exciting progress has been made in the production of medicinal natural products, including artemisinic acid, taxadiene, and opioids, through synthetic biology[11–16]. Therefore, synthetic biology

provides an opportunity to address the resource shortage of bergenin.

Although bergenin has been isolated for nearly 100 years, its biosynthetic pathway remains unclear. Structurally, bergenin is the lactone product of 4-*O*-methyl gallic acid 2-*C*-β-D-glycoside (4-OMGA-Glc, **4**), which is likely to be generated by regio- and/or stereoselective *C*-glycosylation and *O*-methylation of gallic acid (GA, **2**) (Fig. 1a). A $^{14}$*C*-glucose incorporation experiment in *Saxifraga stolonifera* leaves also indicated that GA is a glucosyl acceptor in bergenin biosynthesis[17]. GA is produced by a metabolic branch of the shikimic acid metabolic pathway that occurs widely in plants and microorganisms. The biosynthetic pathway of GA has been clarified, and considerable progress has been made in the biosynthesis of GA[18–20]. However, the biosynthetic pathway of bergenin from GA has not been characterized, especially for 4-*O*-methyltransferase (4-OMT) and 2-*C*-glycosyltransferase (2-CGT) (Fig. 1a). Due to the unique structure of bergenin, it is difficult to utilize known genetic elements for the specific *C*-

[1]Key Laboratory of Ecology and Environment in Minority Areas (National Ethnic Affairs Commission), College of Life and Environmental Sciences, Minzu University of China, Beijing, China. [2]State Key Laboratory of Bioactive Substance and Function of Natural Medicines, Institute of Materia Medica, Chinese Academy of Medical Sciences & Peking Union Medical College, Beijing, China. [3]CAMS Key Laboratory of Enzyme and Biocatalysis of Natural Drugs, Beijing, China. [4]NHC Key Laboratory of Biosynthesis of Natural Products, Beijing, China. [5]These authors contributed equally: Ruiqi Yan, Binghan Xie, Kebo Xie, Qi Liu. ✉e-mail: keboxie@imm.ac.cn; jgdai@imm.ac.cn; yanglin72@muc.edu.cn

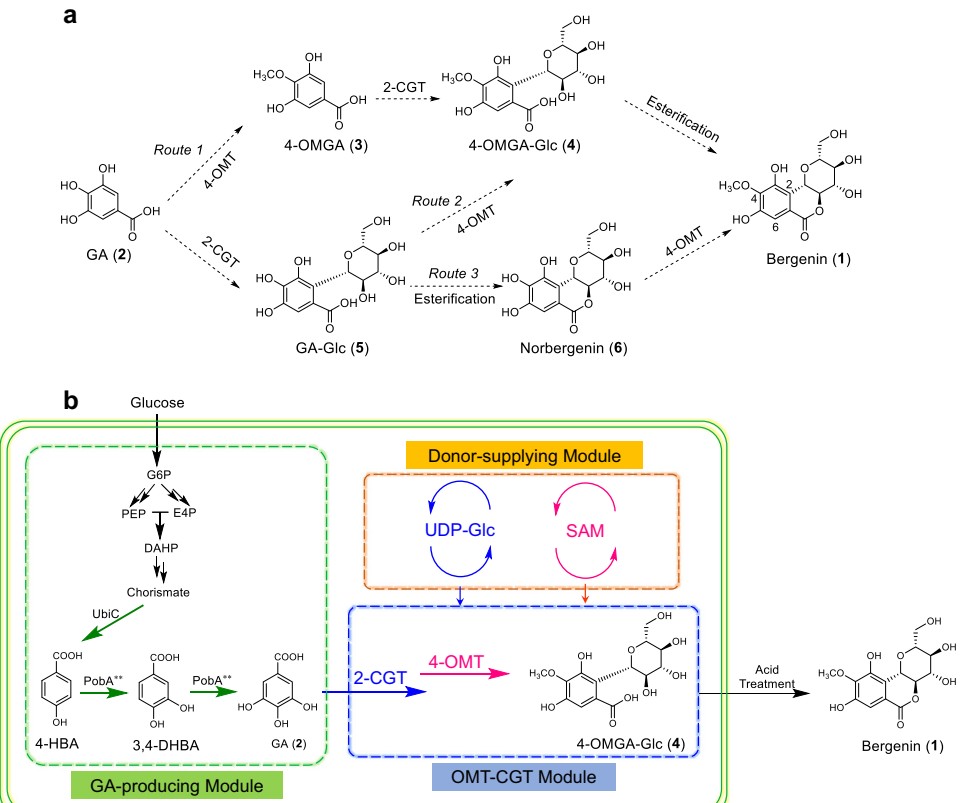

**Fig. 1 | The biosynthetic pathway of bergenin. a** Proposed biosynthetic pathway from GA (**2**) to bergenin (**1**) in *A. japonica*. Dashed arrows represent the unclear steps. **b** The designed de novo artificial biosynthesis of bergenin from glucose in *Escherichia coli*. The reconstructed biosynthetic pathway mainly contains GA-producing module, OMT-CGT module and donor-supplying module. GA, gallic acid (**2**); 4-OMGA, 4-*O*-methyl gallic acid (**3**); 4-OMGA-Glc, 4-*O*-methyl gallic acid 2-*C*-β-D-glycoside (**4**); GA-Glc, gallic acid 2-*C*-β-D-glycoside (**5**); G6P, glucose-6-phosphate; PEP, phosphoenolpyruvate; E4P, erythrose 4-phosphate; DAHP, 3-deoxy-D-arabino-heptulosonate-7-phosphate; 4-HBA, 4-hydroxybenzoic acid; 3,4-DHBA, 3,4-dihydroxybenzoic acid; UbiC, chorismate lyase; PobA**, *p*-hydroxybenzoate hydroxylase with Y385F and T294A mutation; 2-CGT, 2-*C*-glycosyltransferase; 4-OMT, 4-*O*-methyltransferase; UDP-Glc, uridine diphosphate glucose; SAM, *S*-adenosyl methionine.

glycosylation and *O*-methylation in its biosynthesis. For *C*-glycosylation, only a small number of CGTs have been found and most of them are flavonoid CGTs, of which the substrate spectra are relatively narrow[21–32]. GA and its analog 4-*O*-methyl gallic acid (4-OMGA, **3**), the putative aglycone substrates of 2-CGT in bergenin biosynthesis, have the acyl 3,4,5-pyrogallol structure, which is different from the substrates of reported CGTs. Typical reported CGTs including OsCGT[21], FeCGTb[22], UGT708D1[23], TcCGT1[24], PlUGT43[25], MiCGT[26], and AbCGT[27] were further selected to investigate their *C*-glycosylation activity toward substrates with the acyl 3,4,5-pyrogallol structure, and no 2-*C*-glycosylation activity was detected (Supplementary Figs. 1 and 2). For *O*-methylation, a highly regioselective 4-OMT with high efficiency in methylating the C4-OH of GA and its derivatives is vital for the biosynthesis of bergenin but has not been reported until now. Therefore, identifying the key CGT(s) and OMT(s) involved in the biosynthesis of bergenin and its analogs is important for elucidating the biosynthetic pathway of these rare bioactive *C*-glycosides and is necessary for accomplishing the de novo biosynthesis of bergenin to overcome its resource shortages (Fig. 1b).

In this work, we elucidate the biosynthetic pathway of bergenin by identifying key specific 2-CGTs (AjCGT1 and AjCGT2) and highly regioselective 4-OMTs (AjOMT2 and AjOMT3) from *A. japonica*. The 2-CGTs show high catalytic activity against GA (**2**) and 4-OMGA (**3**), generating 2-*C*-β-D-glycosides (**5** and **4**). In addition, the highly regioselective 4-OMTs exhibit 4-*O*-methylation activity toward GA (**2**) and norbergenin (**6**) to yield the corresponding 4-*O*-methylated products (**3** and **1**). We reconstruct the biosynthetic pathway in *Escherichia coli*

and achieve the de novo biosynthesis of 4-OMGA-Glc, which is conveniently esterified to bergenin by acid treatment in situ with a titer of 1.41 g L⁻¹. This work provides an alternative to plant extracts for large-scale production of bergenin.

## Results and discussion

### Unraveling the biosynthetic pathway of bergenin

The biosynthetic pathway from GA (**2**) to bergenin (**1**) is unknown. GA may be catalyzed successively by a regioselective 4-*O*-methyltransferase (4-OMT), a regio- and stereoselective *C*-glycosyltransferase (2-CGT), and an enzyme for intramolecular esterification (Fig. 1a). Due to the unique structure of bergenin, no known genetic elements are available for specific *C*-glycosylation and *O*-methylation reactions for the reconstruction of artificial biosynthetic pathways for bergenin. Therefore, we attempted to elucidate the biosynthetic pathway of bergenin by identifying two key enzymes.

Bergenin was detected throughout the whole *A. japonica* plant; however, the contents in different organs (leaf, stem, and rhizome) varied, and the highest content was found in the leaf (Fig. 2a, b and Supplementary Table 1). According to the differential transcriptome data of the leaf, stem, and rhizome, 85 candidate glycosyltransferase genes were selected. Subsequently, phylogenetic analysis of these glycosyltransferase candidates and the previously reported CGTs indicated that two glycosyltransferase candidates (CL7566-1 and Unigene14257, named AjCGT1 and AjCGT2, respectively) and the typical plant CGTs were grouped into the same clade (Supplementary Fig. 3). Moreover, the expression levels of *AjCGT1* and *AjCGT2* in different

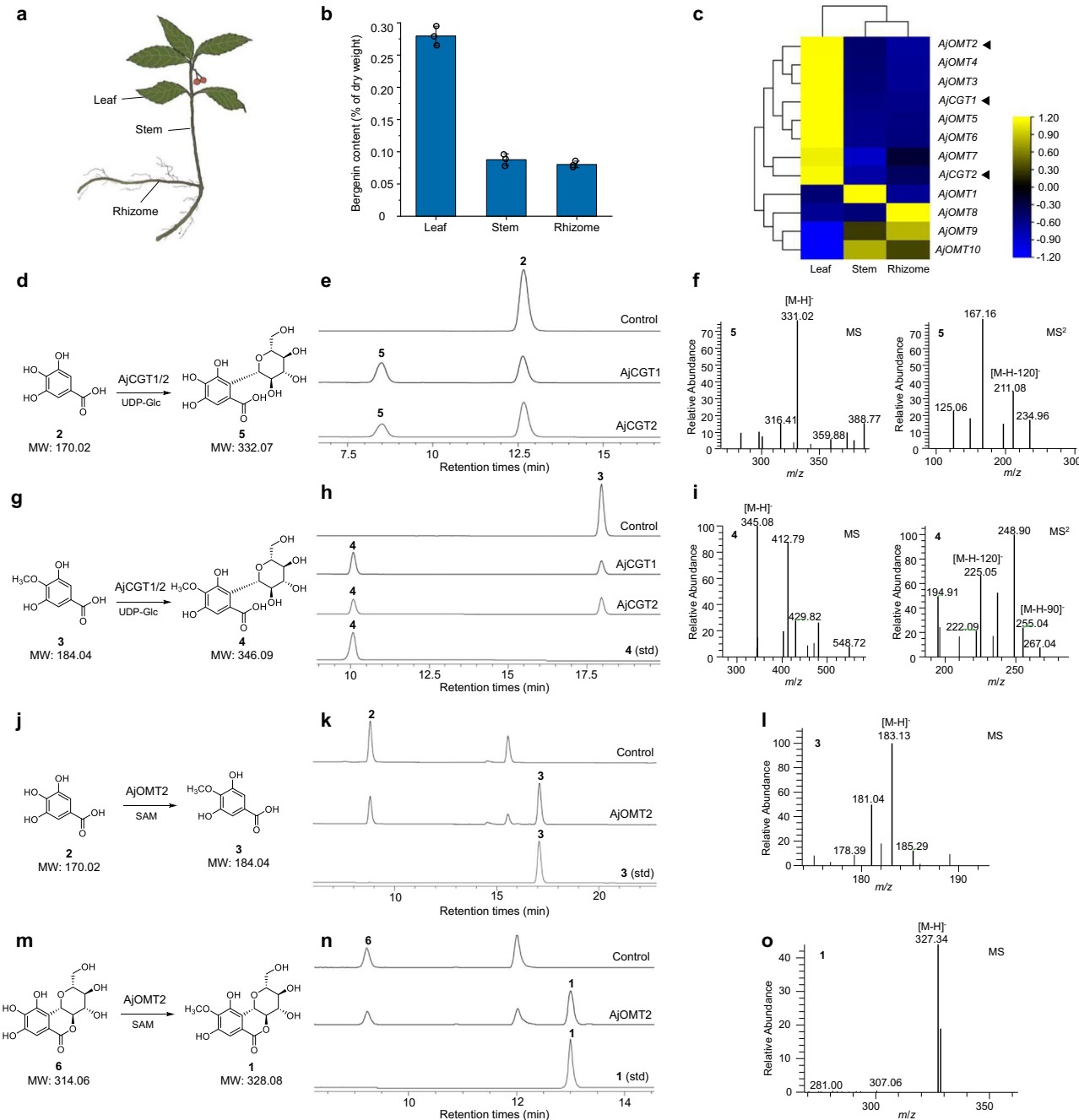

**Fig. 2 | Functional identification of the 2-CGTs and 4-OMTs involved in the biosynthesis of bergenin. a** The organs (leaf, stem, and rhizome) of *A. japonica* were used to analyze the contents of bergenin (**1**) and transcriptome sequencing. **b** The contents of bergenin (**1**) in different organs, all data represent the means of three parallel experiments and error bars show standard deviation. **c** Expression heat-map of two candidate CGT genes and ten OMT candidate genes (*AjOMT1–10*) in different organs. **d**–**f** The reactions catalyzed by AjCGT1 and AjCGT2 with GA (**2**) and UDP-Glc as substrates, and the corresponding HPLC-MS/MS[2] spectra. **g**–**i** The reactions catalyzed by AjCGT1 and AjCGT2 with 4-OMGA (**3**) and UDP-Glc as substrates, and the corresponding HPLC-MS/MS[2] spectra. **j**–**l** The reactions catalyzed by AjOMT2 with GA (**2**) and SAM as substrates, and the corresponding HPLC-MS spectra. **m**–**o** The reactions catalyzed by AjOMT2 with norbergenin (**6**) and SAM as substrates, and the corresponding HPLC-MS spectra. Source data are provided as a Source data file.

organs were strongly correlated with the content of bergenin in the corresponding organs (Fig. 2c and Supplementary Tables 1 and 2); therefore, these two genes were chosen as CGT candidates for further functional identification.

The full-length cDNAs of *AjCGT1* and *AjCGT2* were cloned and expressed in the *E. coli Trans*etta (DE3) strain (Supplementary Data 1 and Supplementary Fig. 4), respectively. The catalytic activity of recombinant AjCGT1 and AjCGT2 was assayed with the putative substrates GA (**2**) and 4-OMGA (**3**) as well as uridine diphosphate glucose (UDP-Glc).

HPLC-MS analysis of the reactions showed that both AjCGT1 and AjCGT2 exhibited glucosylation activity toward **2** and **3**, generating products **5** and **4**, respectively, with the same retention time as the standards (Fig. 2d–i). MS analysis revealed that the [M−H]− ions of **5** and **4** were 162 amu greater than those of **2** and **3**, indicating that **5** and **4** were the corresponding mono-glucosylated products (Fig. 2f, i). The characteristic fragment ions [M−H−120]− and [M−H−90]− of **5** and **4** suggested that both were *C*-glucosylated products (Fig. 2f, i). Gallic acids are unstable in buffer solutions and easily undergo nonspecific binding with

proteins[33,34] (Supplementary Figs. 5–7). Thus, the apparent $K_m$ and $k_{cat}$ were distorted, and the specific activities of the AjCGTs were measured (Supplementary Table 3). Compared with **2**, both AjCGT1 and AjCGT2 showed greater catalytic activity toward **3**, suggesting that **3** is more likely the native substrate (Supplementary Table 3). Therefore, AjCGT1 (UGT708AL1) and AjCGT2 (UGT708AL2) were tentatively identified as CGTs responsible for the specific 2-*C*-glucosylation involved in the biosynthesis of bergenin. AjCGT1 also has *C*-glycosylation activity toward common CGT substrates with 2,4,6-trihydroxyacetophenone-like structures (Supplementary Figs. 8–12), while the seven tested CGTs showed no activity toward **2** and **3**, as mentioned previously (Supplementary Figs. 1 and 2). Therefore, in contrast to other reported plant CGTs, the two CGTs displayed the ability to introduce *C*-sugars into the aromatic ring with acyl 3,4,5-pyrogallol structures.

To further investigate the mechanism by which AjCGT1 catalyzes gallic acid synthesis, a structural model of AjCGT1 was obtained according to the crystal structure of SbCGTa (PDB code 6LG0)[28]. GA and 4-OMGA were docked into AjCGT1, respectively (Supplementary Fig. 13). Fourteen candidate active sites located adjacent to the substrate binding pocket were selected. Alanine scanning assays revealed that 10 mutants (H23A, H27A, W92A, T142A, F147A, W151A, F188A, H367A, G368A, D369A, and Q370A) lost their activity. The key sites in the binding pocket ensure that gallic acid is in the proper orientation. The conserved His23 is generally considered a vital active site for the specific deprotonation of the hydroxyl group of the substrate, which makes it a nucleophile that attacks the sugar donor[35], and the AjCGT1-H23A mutant completely lost its activity. However, two other basic candidate sites, His27 and W92, were found near the highly conserved His23 (Supplementary Fig. 14). In particular, the *C*-glycosylation activity of W92A toward gallic acid was completely lost, while that toward phloretin remained at approximately 15% (Supplementary Fig. 14). Therefore, the His27 and W92 basic sites might also act as general bases to deprotonate the hydroxyl group of gallic acid, helping His23 complete the *C*-glycosylation of gallic acid.

Owing to the relatively higher conversion rates of AjCGT1 (45.8% for **2**, 61.5% for **3**) than those of AjCGT2 (29.8% for **2** and 47.0% for **3**) with GA (**2**) and 4-OMGA (**3**), AjCGT1 was selected as the target 2-CGT for further biochemical property investigations and engineered strain construction (Supplementary Figs. 15 and 16). Under pH 8.0, AjCGT1 exhibited the optimum activities at 35 °C and 40 °C for **2** and **3**, respectively, and were independent of metal ions (Supplementary Figs. 17 and 18). Kinetic analysis demonstrated that AjCGT1 exhibited an apparent $K_m$ value of 265.3 μM for 4-OMGA (**3**). The corresponding $K_{cat}/K_m$ value ($1.2 \times 10^3 \, M^{-1} \, s^{-1}$ for **3**) of AjCGT1 is lower than that of known CGTs, which might be affected by the instability of **3** during the reaction (Supplementary Fig. 19). When the stable substrate phloretin was used, AjCGT1 exhibited relatively high specific activity (Supplementary Table 3). Next, an engineered strain harboring the *AjCGT1* gene was constructed and converted exogenously added **2** and **3** to **5** and **4**, respectively (Supplementary Figs. 20 and 21). These results indicated that *AjCGT1* was compatible with *E. coli* and can be used for reconstructing engineered strains for the de novo production of bergenin precursor **4**.

Ten candidate methyltransferase genes (*AjOMT1–10*) for polyphenol biosynthesis with relatively high expression levels were screened from the differential transcriptome data of *A. japonica* (Supplementary Table 4). Among these genes, the expression levels of six genes (*AjOMT2–7*) were positively correlated with the levels of bergenin in the corresponding organs and the bait gene expression levels of *AjCGT1* and *AjCGT2* (Fig. 2c). Considering that genes related to the biosynthetic pathway of secondary metabolites in plants are usually co-ordinately regulated[36], AjOMT2–7 and representative OMTs were selected for phylogenetic analysis (Supplementary Fig. 22). The results showed that AjOMT2 and AjOMT3 were clustered into the clade of caffeoyl-CoA OMTs, which tended to decorate one of the adjacent hydroxyl groups on the aromatic rings of the substrate[37,38]. Moreover, AjOMT2 and AjOMT3 were clustered close to AtCCoAOMT7, which displayed unusual *para*-methylation activity[39]. These findings suggest that AjOMT2 and AjOMT3 are involved in the biosynthesis of bergenin.

Accordingly, *AjOMT2* and *AjOMT3* were cloned from the cDNA of *A. japonica* and heterologously expressed in *E. coli*, and the recombinant enzymes were purified by Ni²⁺-NTA affinity chromatography (Supplementary Data 1 and Supplementary Fig. 23)[40]. To characterize the function of AjOMTs, two putative precursors (**2** and **6**) involved in bergenin biosynthesis were used as methyl acceptors, and *S*-adenosyl-L-methionine (SAM) was used as the methyl donor. For AjOMT2, a product showing an identical retention time and molecular ion peak at *m/z* 183.13 ([M–H]⁻) to those of standard **3** (Fig. 2j–l) appeared in the reaction mixture of **2** with SAM. This demonstrated that recombinant AjOMT2 performed regiospecific methylation at the 4-OH of **2** to produce **3** with a conversion rate of 62.4%. In addition, HPLC-MS analysis revealed that AjOMT2 can also methylate **6** to form **1** with a conversion rate of 72.2% (Fig. 2m–o). The enzymatic assay showed that AjOMT3 also exhibited 4-*O*-methylation activity toward **2** and **6** in vitro; however, the catalytic activity was lower than that of AjOMT2 (Supplementary Figs. 24 and 25). In addition, AjOMT4−7 exhibited much lower or no catalytic activity (Supplementary Figs. 26 and 27). Therefore, both AjOMT2 and AjOMT3 might be regioselective 4-OMTs involved in the biosynthesis of bergenin, and AjOMT2 was selected for further biochemical property investigations and engineered strain construction.

Biochemical property investigations revealed that the optimum temperature and pH of AjOMT2 were 40 °C and 7.0, respectively. This enzyme was dependent on metal ions, and Mg²⁺ and Mn²⁺ could enhance its activity toward **2** and **6** (Supplementary Figs. 28 and 29). AjOMT2 showed methylation activity to **2** and **6** with similar conversion rates while kinetic properties demonstrated that AjOMT2 had apparent $K_m$ values of 72.0 μM and 493.6 μM for **2** and **6**, respectively (Supplementary Fig. 30). These results implied that the native substrate of AjOMT2 might be **2** rather than **6** and that 4-*O*-methylation may occur before 2-*C*-glycosylation. In comparison with other similar OMTs, AjOMT2 had a moderate $K_{cat}/K_m$ value ($250 \, M^{-1} \, s^{-1}$) (Supplementary Table 5). Moreover, the engineered *E. coli* strain harboring *AjOMT2* also exhibited 4-*O*-methylation activity when **2** and **6** were exogenously added (Supplementary Figs. 31 and 32). Thus, AjOMT2 is an efficient and highly regioselective methyltransferase compatible with *E. coli* for artificial biosynthetic pathway construction.

Therefore, the specific 2-CGTs together with the regio-selective 4-OMTs accomplished the enzymatic synthesis of the specific *C*-glycoside 4-OMGA-Glc (**4**) from GA (**2**). Furthermore, the target bioactive product bergenin was conveniently obtained by acid treatment of 4-OMGA-Glc (**4**) (Supplementary Fig. 33). Thus, the biosynthetic pathway of bergenin was elucidated by identifying key enzymes and providing efficient genetic elements for the biosynthesis of bergenin in cell factories.

### Reconstructing the biosynthetic pathway of 4-OMGA-Glc from GA

To reconstruct the biosynthetic pathway of the bergenin direct precursor 4-OMGA-Glc (**4**), the OMT-CGT (4-*O*-methyltransferase and 2-*C*-glycosyltransferase) module was designed by functional and effective coexpression of AjOMT2 and AjCGT1 in *E. coli*. Plasmids with different copy numbers were used to carry *AjOMT2* and *AjCGT1*, different combination models of these recombinant plasmids were designed, and their catalytic efficiency and fitness were tested in vivo. *AjOMT2* and *AjCGT1* were cocloned and inserted into the high-copy number plasmid pETDuet-1, middle-copy number plasmid pCDFDuet-1 and low-copy number plasmid pACYCDuet-1, respectively, which were subsequently transformed into *E. coli* BL21 (DE3), leading to three engineered strains, E1−E3 (Supplementary Table 6). In the presence of

endogenous UDP-Glc and SAM, **4** was detected in the culture media of all the strains after an additional 24 h of incubation after the addition of exogenous **2** at a final concentration of 0.4 mM through whole-cell biocatalysis (Supplementary Fig. 34). Moreover, the GA in the culture media of strain E1 harboring plasmid pET-*AjOMT2-AjCGT1* and strain E3 carrying plasmid pACYC-*AjOMT2-AjCGT1* were completely consumed, suggesting the high efficiency and fitness of AjCGT1 and AjOMT2. Therefore, the combination of AjCGT1 with AjOMT2 leads to the successful synthesis of the specific *C*-glycoside 4-OMGA-Glc (**4**) from GA (**2**).

## De novo biosynthesis of the bergenin precursor 4-OMGA-Glc

To accomplish the de novo biosynthesis of the bergenin precursor 4-OMGA-Glc (**4**), we attempted to combine the exogenous biosynthetic pathway of GA (**2**) (GA-producing module) with AjOMT2 and AjCGT1 (OMT-CGT module) using *E. coli* as the chassis cell. GA is an important precursor of the biosynthetic pathway of bergenin; however, unengineered *E. coli* cannot supply GA. According to the designed artificial biosynthetic pathway of GA in *E. coli* (Fig. 1b), one of the vital factors is the lack of natural hydroxylase catalyzing 3,4-dihydroxybenzoic acid (3,4-DHBA) into GA. It has been reported that PobA-Y385F/T294A (PobA**), a mutant of *p*-hydroxybenzoate hydroxylase (PobA) from *Pseudomonas fluorescens*, exhibits high hydroxylase activity toward 4-hydroxybenzoic acid (4-HBA) and 3,4-DHBA to form GA[20]. Moreover, chorismate lyase (UbiC) was demonstrated to enhance the transformation of chorismate to 4-HBA with high catalytic activity in *E. coli*[41]. Therefore, for efficient de novo biosynthesis of GA, *PobA***, and *UbiC* were used to construct the GA biosynthetic module (Fig. 3a). Exogenous *PobA*** and endogenous *UbiC* were constructed from the above three plasmids with different copy numbers. Subsequently, the plasmids were transformed into *E. coli* to create three engineered strains S1−S3, which contained the plasmids pET-*PobA***-*UbiC*, pCDF-*PobA***-*UbiC*, and pACYC-*PobA***-*UbiC*, respectively (Supplementary Table 6). GA was detected in the culture supernatant of the three engineered strains after 24 h of cultivation, respectively (Supplementary Fig. 35). The yield of GA in strain S1 reached 619 mg L$^{-1}$, 26% and 59% higher than those in strains S2 and S3, respectively. Thus, a highly GA-producing pathway (GA-producing module) was constructed.

Generally, increasing the copy numbers of functional genes is a strategy used to increase the level of gene expression; however, the process might impose greater physiological burdens on hosts and even decrease the yields of target products[42,43]. In this context, to determine the best combination mode of *AjOMT2-AjCGT1* and *PobA***-*UbiC*, three plasmids carrying *AjOMT2-AjCGT1* with different copy numbers were introduced into the three GA-producing strains S1−S3 (carrying *PobA***-*UbiC*), generating six strains, A1−A6, respectively (Table 1 and Fig. 3a). 4-OMGA-Glc (**4**) was detected in the fermentation broth supernatants of all six strains after expression was induced for 24 h (Supplementary Fig. 36), and the titer of 4-OMGA-Glc in the engineered strain A3 carrying the pCDF-*PobA***-*UbiC* and pET-*AjOMT2-AjCGT1* plasmids was the highest, reaching 418 mg L$^{-1}$ at 48 h in a shake flask (Fig. 3b and Supplementary Fig. 37). Moreover, intermediate products during fermentation accumulated and varied among the different engineered strains (Fig. 3b). These findings suggested that when multiple genes were coexpressed on two different plasmids, the degree of compatibility between plasmids and genes strongly affected the yield of the target product. Overall, strains A1−A6 could produce 4-OMGA-Glc from the simple carbon source glucose; thus, the de novo biosynthetic pathway of 4-OMGA-Glc was reconstructed.

## Optimizing the biosynthetic pathway of the bergenin precursor 4-OMGA-Glc

As a large amount of GA accumulates in the culture medium of strain A3 (Fig. 3b), both the efficiency of the OMT-CGT module and the

donor-supplying module should be further optimized to increase the yield of 4-OMGA-Glc based on the strain A3.

Semi-rational design guided by structural information is considered as an efficient and economical strategy for protein engineering to improve the catalytic activity of target enzymes[20,44,45]. Therefore, to further improve the catalytic efficiency of the OMT-CGT module, a strategy of semirational design was employed to increase the activity of AjOMT2 and AjCGT1 (Fig. 4). A structural model of AjOMT2 was generated using sorghum caffeoyl-CoA *O*-methyltransferase (CCoAOMT, PDB code 5KVA) as the template in complex with SAM[46]. GA and GA-Glc were docked into AjOMT2, respectively (Fig. 4a and Supplementary Fig. 38). Seven residues (A156, D157, K158, E159, N160, W185, and Y203) located adjacent (4 Å) to the substrate binding pocket of AjOMT2 were found, and six residues, excluding A156, were mutated to alanine. The K158A mutant almost completely lost its activity, revealing that K158 might act as a catalytic base and deprotonate the reactive hydroxyl group of gallic acid, which is similar to the reported catalytic mechanism[46] (Supplementary Fig. 39). The catalytic activities of the mutants were further investigated in vivo through combining the wild-type AjCGT1. The strain harboring *AjOMT2-Y203A* and wild-type *AjCGT1* significantly increased the conversion rate of GA to 4-OMGA-Glc (Fig. 4b). Subsequently, site-directed saturation mutagenesis of the Y203 residue was performed, and the Y203T variant exhibited the highest activity, which was 31% higher than that of wild-type AjOMT2 (Fig. 4c).

As none of the mutants exhibited increased catalytic activity according to the strategy of the structural model, molecular docking, and site mutation of AjCGT1, HotSpot Wizard was used to engineer AjCGT1 (Fig. 4d). The HotSpot Wizard is a web server focused on automated prediction of hotspot residues for mutagenesis that are likely to alter enzyme activity[47]. Then, the candidate sites V328, E329, E331, and A332 were selected and further mutated to V328A, E329A, E331A, and A332E/D/F/K/Q/V/Y, respectively, with the guidance of HotSpot Wizard (Fig. 4e). Among these mutants, AjCGT1-A332F exhibited the highest catalytic activity (138% that of the wild-type AjCGT1) (Fig. 4e).

Next, the AjOMT2-Y203T (designated AjOMT2*) and AjCGT1-A332F (designated AjCGT1*) mutants were coexpressed in engineered *E. coli* strains (M1−M3) (Fig. 4f). Surprisingly, the catalytic activity of the engineered strain harboring the plasmid pET-*AjOMT2*-AjCGT1** (M3) significantly decreased by 29% compared with that of the strain harboring wild-type *AjOMT2-AjCGT1* (E1) (Fig. 4g). However, the combinations of *AjOMT2*-AjCGT1* (M1) and *AjOMT2-AjCGT1** (M2) could improve the conversion rate of GA to 4-OMGA-Glc. Among the engineered strains, the M2 strain carrying pET-*AjOMT2-AjCGT1** exhibited the highest catalytic activity, of which the conversion rate was 38% higher than that of the strain harboring wild-type *AjOMT2-AjCGT1* (Fig. 4g).

In general, codon optimization is an effective way to improve the protein expression level in *E. coli* strains[48]. Therefore, AjOMT2, AjCGT1, AjOMT2* and AjCGT1* were codon-optimized based on the codon preference of *E. coli*. In total, five co-expression plasmids were designed, constructed and transformed into *E. coli* BL21 (DE3) to generate five strains, Y1−Y5, respectively (Fig. 4h and Supplementary Table 6). As expected, the catalytic efficiency of strain Y5 harboring *AjOMT2^opt* and *AjCGT1*^opt* (codon-optimized strain M2) increased by 44% (Fig. 4i). However, strain Y4 harboring codon-optimized *AjOMT2*^opt* and *AjCGT1^opt* (codon-optimized strain M1) showed the highest catalytic activity, of which the conversion rate increased by 65% compared with that of strain M2 (Fig. 4i). Hence, strain Y4 harboring pET-*AjOMT2*^opt-AjCGT1^opt* was used to construct the next generation of the engineered strain.

In the biosynthetic pathway of 4-OMGA-Glc (**4**), the donors (SAM and UDP-Glc) play crucial roles, and their adequate supply could help direct carbon flux toward target products[49]. According to the above

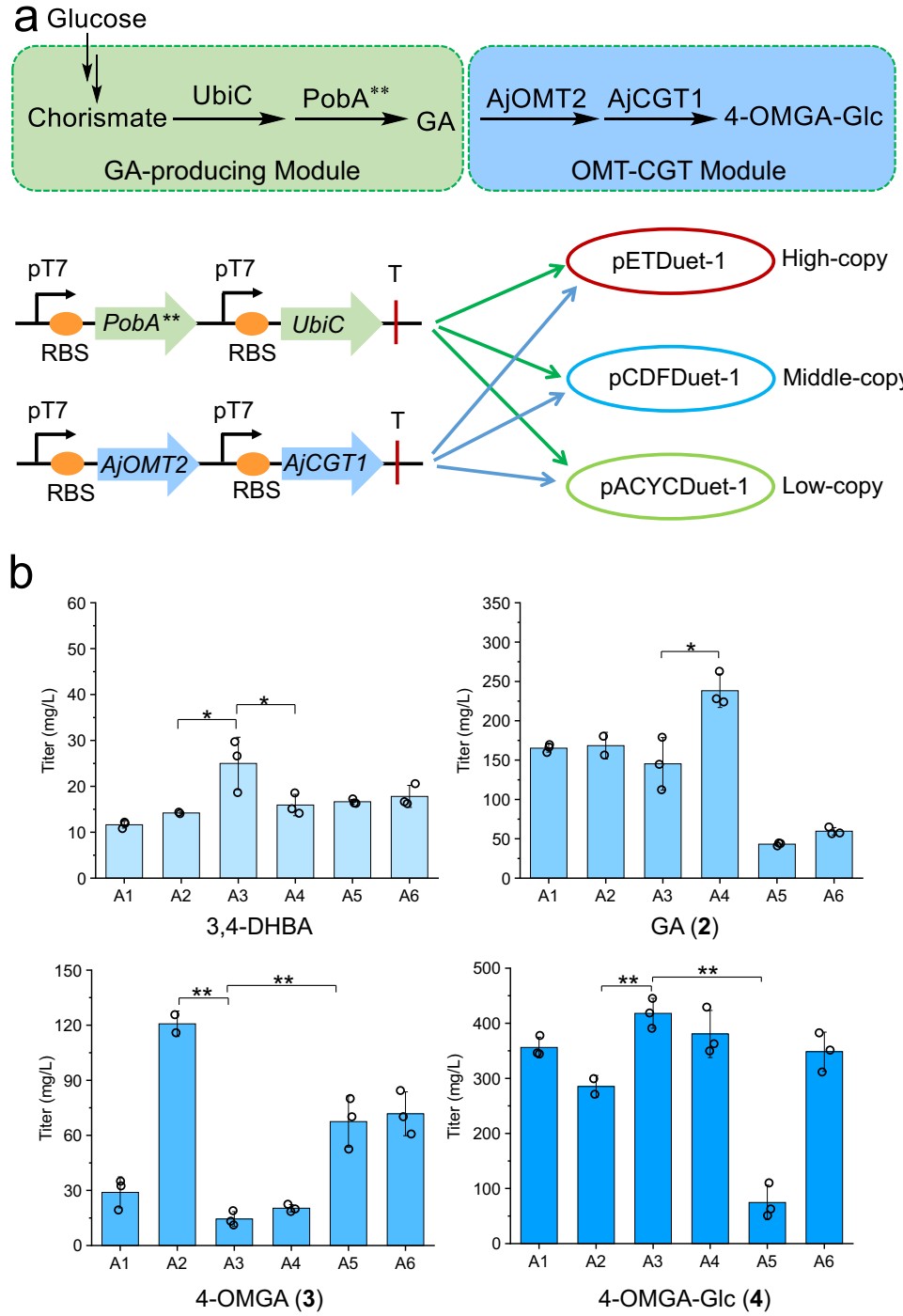

**Fig. 3 | De novo biosynthesis of the bergenin precursor 4-OMGA-Glc from glucose in *E. coli*. a** The construction of the GA-producing module and OMT-CGT module using different plasmids. **b** Evaluation of the engineered strains A1–A6 for the titers of 4-OMGA-Glc (**4**) and metabolic intermediates in shake flask fermentation for 48 h. UbiC, chorismate lyase; PobA**, *p*-hydroxybenzoate hydroxylase with Y385F and T294A mutation; pT7, T7 promoter; RBS, ribosome binding site; T, terminator. All data represent the means of three parallel experiments and error bars show standard deviation. Statistical analysis was performed by using the Student's *t* test. *P* value for each comparison from left to right then top to bottom in **b**): 0.0408 (*), 0.0475 (*), 0.0106 (*), 0.0047 (**), 0.0081 (**), 0.0048 (**), 0.0001 (**). Source data are provided as a Source data file.

results, the accumulation of GA in strain A3 suggested that the supply of endogenous SAM and UDP-Glc was insufficient. Therefore, to further increase the supply of methyl and sugar donors, five key enzymes involved in the biosynthesis of SAM (5′-methylthioadenosine/*S*-adenosylhomocysteine nucleosidase, mtn; *S*-ribosylhomocysteine lyase, luxS; methionine adenosyltransferase, metK)[49] and UDP-Glc

(phosphoglucomutase, pgm; glucose-1-phosphate uridyltransferase, galU)[50] were utilized for engineering *E. coli* (Fig. 5a, b). These five endogenous enzyme-encoding genes individually or in combination were overexpressed together with *PobA** and *UbiC* in strain Y4, generating an additional three engineered strains, D3–D5 (Fig. 5c). Strain D1 contained the plasmids pCDF-*PobA***-*UbiC* and pETD-*AjOMT2**opt*-

**Table 1 | Engineered *E. coli* strains A1–A6**

| Recombinant plasmids | A1 | A2 | A3 | A4 | A5 | A6 |
|---|---|---|---|---|---|---|
| pETDuet-*PobA**-UbiC* | + | + | – | – | – | – |
| pCDFDuet-*PobA**-UbiC* | – | – | + | + | – | – |
| pACYCDuet-*PobA**-UbiC* | – | – | – | – | + | + |
| pETDuet-*AjOMT2-AjCGT1* | – | – | + | – | + | – |
| pCDFDuet-*AjOMT2-AjCGT1* | – | + | – | – | – | + |
| pACYCDuet-*AjOMT2-AjCGT1* | + | – | – | + | – | – |

Strains A1–A6 harbor recombinant plasmids with different combinations of *UbiC*, *PobA***, *AjOMT2*, and *AjCGT1*. +: harboring plasmids; –: not harboring plasmids.

*AjCGT1^opt^*, and strain D2 was constructed by introducing pACYCDuet-1-blank into strain D1 (Fig. 5c). Strain A3 was also used as a control to compare the effect of this optimization.

Compared with that in strain A3, the yield of 4-OMGA-Glc in strain D1 increased by 34%, reaching 560 mg L$^{-1}$ after fermentation for 48 h (Fig. 5d). Surprisingly, when a third empty plasmid was introduced into the 4-OMGA-Glc-producing strain D1 to obtain strain D2, the yield of the target product decreased by 26% in 48 h (Fig. 5d). Introducing extra plasmids might increase the metabolic burden and stress to host cells, thereby reducing the accumulation of target products. Therefore, the metabolic fluxes and protein expression levels in biosynthetic pathways containing multiple genes should be accurately regulated to maximize the 4-OMGA-Glc titer. When *metK* was overexpressed in strain D3, the titer of 4-OMGA-Glc decreased by 6% compared with that in strain D2 after 48 h (Fig. 5d). This result suggested that direct overexpression of *metK* does not improve the SAM supply, probably due to the complicated regulation of the SAM biosynthetic pathway in *E. coli*. Reasonably, when *mtn* and *luxS* were overexpressed in strain D4, the titer of the product 4-OMGA-Glc was increased by 46% (reaching 610 mg L$^{-1}$) compared with that in strain A3 after 48 h (Fig. 5d). These findings indicated that constructing a SAM regeneration system to increase SAM availability is a reliable strategy for increasing the yield of 4-OMGA-Glc.

For the sugar donor, the biosynthetic pathway of UDP-Glc commonly occurs in organisms; however, further increasing the UDP-Glc supply might increase the yield of the corresponding glycosides. Therefore, two key enzymes (pgm and galU)[50] for its biosynthesis were overexpressed in strain D5. Surprisingly, the titer of the product 4-OMGA-Glc decreased by 27% compared with that of strain A3 after 48 h (Fig. 5d), probably due to their poor fitness. It was speculated that the overexpression of *pgm* and *galU* caused carbon flux to UDP-Glc, which might affect the growth of *E. coli* cells. Future work involving the overexpression of sucrose synthase (SuSy) might be an alternative method for increasing the supply of UDP-Glc[45].

Overall, by optimizing the OMT-CGT module, the 4-OMGA-Glc titer of strain D1 increased to 560 mg L$^{-1}$ at 48 h. With further optimization of the donor-supplying module, the highest amount of 4-OMGA-Glc reached 610 mg L$^{-1}$ at 48 h in the resultant strain D4, which was 46% higher than that of the starting strain A3.

### Fed-batch production of 4-OMGA-Glc in a bioreactor
To evaluate the scale-up potential of D4 for 4-OMGA-Glc production, fed-batch experiments were carried out in a 3-L bioreactor. After 24 h of fermentation, the upstream intermediate products 3,4-DHBA and GA started to be produced and accumulated in maximum yields of 314 mg L$^{-1}$ and 70 mg L$^{-1}$, respectively. After that, the yield of the target product 4-OMGA-Glc gradually increased, reaching a maximum titer of 1.49 g L$^{-1}$ after 60 h of fermentation (Fig. 6). In previous studies, great progress was made in the de novo production of plant-derived

phenolic *C*-glycosides from glucose. The concentrations of puerarin (daidzein 8-*C*-glucoside), isovitexin (apigenin 6-*C*-glucoside), and apigenin di-*C*-arabinoside produced by engineered yeast reached 72.8 mg L$^{-1}$, 206 mg L$^{-1}$ and 113.16 mg L$^{-1}$, respectively[16,51,52]. A total of 93.9 mg L$^{-1}$ vitexin (apigenin 8-*C*-glucoside) was produced by engineered *E. coli*[53]. In our work, the titer of *C*-glycosides was improved to the gram level per liter.

The desired 4-OMGA-Glc was the main product in the fermentation broth. Although a large amount of GA accumulated during fermentation in shake flasks (Fig. 5d and Supplementary Fig. 40), nearly all the GA in these fed batches was consumed by downstream enzymes (Supplementary Fig. 41). Therefore, after optimization, the biosynthetic genes functionally and efficiently worked together in engineered *E. coli*, with the highest yield of 4-OMGA-Glc ever reported. Moreover, the engineered strain showed great potential for increasing the production of the target *C*-glycoside. After fermentation, when the pH of the broth was adjusted to 0.1 with HCl, 4-OMGA-Glc was completely esterified to bergenin (Fig. 6 and Supplementary Fig. 42) with a yield of 1.41 g L$^{-1}$. Due to the low content of byproducts during fermentation, bergenin was conveniently isolated and purified to meet commercial demand. Therefore, we rebuilt an artificial biosynthetic pathway for bergenin and improved its yield to the gram level per liter, demonstrating great potential for addressing the resource scarcity issue.

In the present work, two key enzymes (AjCGTs and AjOMTs) responsible for the biosynthesis of bergeinin were functionally identified, and the biosynthetic pathway involved was completely elucidated. AjOMT2 and AjOMT3 exhibited highly regioselective 4-*O*-methylation of GA. The two CGTs (AjCGT1 and AjCGT2) were characterized as CGTs of gallic acids, showing strict regio- and stereospecificity for the formation of 2-*C*-β-D-glycosides, different from other reported plant CGTs. Subsequently, the de novo artificial biosynthetic pathway of the bergenin direct precursor 4-OMGA-Glc was reconstructed in *E. coli*. After that, the optimization of the cell factory producing target rare *C*-glycosides was successively and systemically performed, including the GA-producing module, the OMT-CGT module and the donor-supplying module. The yield of the target product was improved, with the highest reported yield of 4-OMGA-Glc to date. Moreover, 4-OMGA-Glc was conveniently transformed to bergenin after the pH of the culture was adjusted in situ. This work illustrates the potential of a bacterial fermentation system to increase the yield of rare *C*-glycosides through metabolic engineering and establishes a method for the biochemical synthesis of bergenin. The efficient and green production of bergenin from glucose lays a foundation for the supply of bergenin.

## Methods
### Plant materials and chemical reagents
*A. japonica* plants (from Xiamen, Fujian Province of China) used in this study were identified by Dr. Lin Yang. Norbergenin (**6**) was isolated from the aerial of *A. japonica* (Supplementary Method 1 and Supplementary Figs. 43 and 44). 4-*O*-methyl gallic acid 2-*C*-β-D-glycoside (**4**) was obtained through bergenin (**1**) hydrolysis (Supplementary Method 2 and Supplementary Figs. 45 and 46). The chemical standards of bergenin (**1**), GA (**2**), 4-OMGA (**3**), and 3,4-DHBA were purchased from Beijing InnoChem Science & Technology Co., Ltd. Uridine 5′-diphosphate glucose (UDP-Glc), *S*-adenosyl-L-methionine (SAM) were purchased from Beijing BG Biotech Co. Ltd. High-performance liquid chromatography (HPLC)-grade acetonitrile or methanol (Thermo Fisher Scientific, United States) was used for HPLC and liquid chromatograph-mass spectrometer (LC-MS) analysis. KOD-Plus DNA Polymerase was purchased from Toyobo Biotech Co., Ltd. Primer synthesizing and DNA sequencing (listed in Supplementary Data 1) was performed by Sangon Biotech Co. Ltd. Polyether defoamer (CS-4809C) was obtained from Sixin North America Inc.

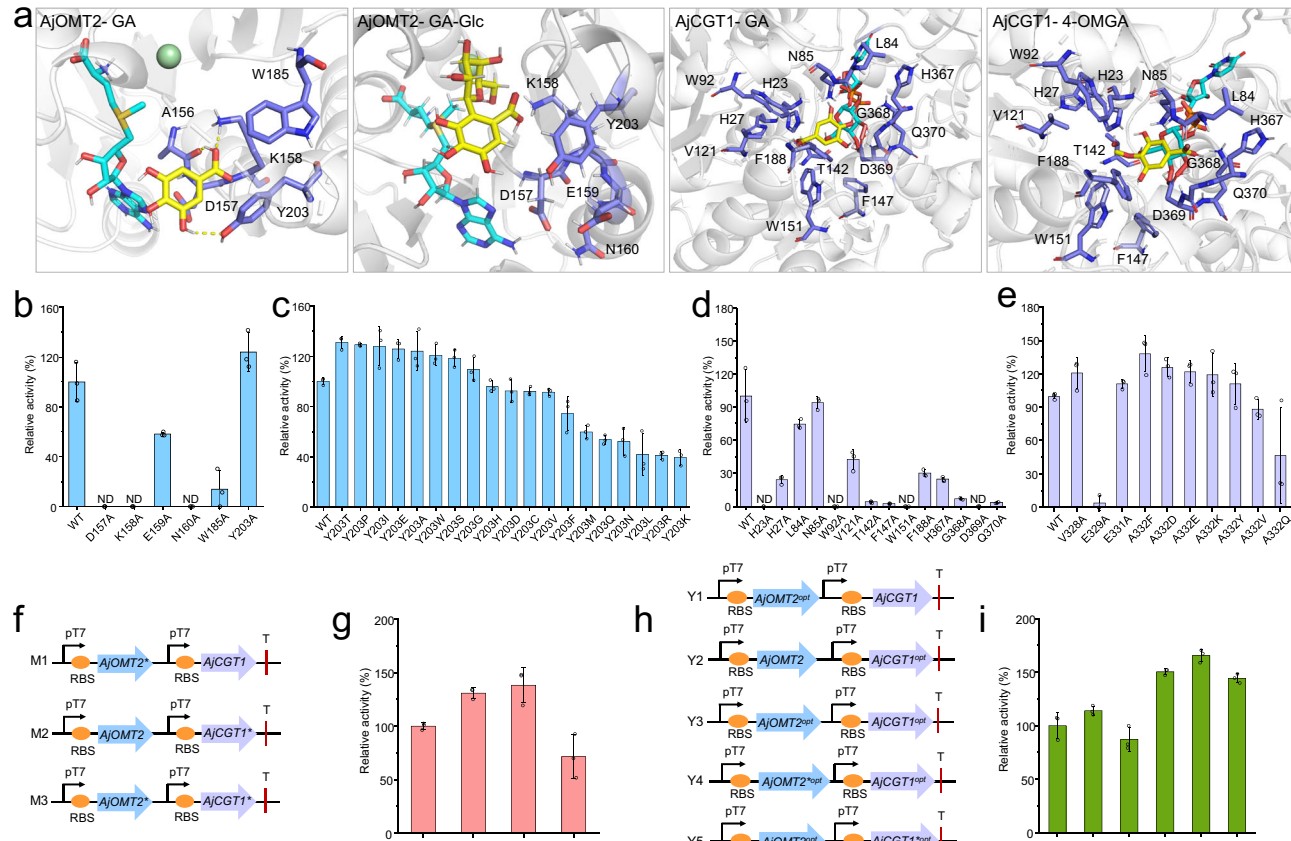

**Fig. 4 | Improving the catalytic efficiency of the OMT-CGT module by semirational design and codon optimization. a** Protein modeling of AjOMT2 and AjCGT1. The protein structures were modeled using CCoAOMT (5KVA) and SbCGTa (6LG0) as templates, respectively. GA and GA-Glc were docked into AjOMT2, while GA and 4-OMGA were docked into AjCGT1 using AutoDock Vina. The candidate sites, acceptors (GA, GA-Glc, and 4-OMGA), and donors (SAM and UDP-Glc) were shown in blue, yellow, and green, respectively. **b**, **c** The in vivo relative catalytic activity of AjOMT2 mutants of alanine scanning and site-saturated mutagenesis of the Y203 site. **d** The in vitro relative activity of AjCGT1 mutants of alanine scanning. **e** The in vivo relative activity of AjCGT1 mutants obtained by Hot-spot analysis.

**f** Engineered strains of *E. coli* harboring different combinations (M1–M3) of *AjOMT2, AjOMT2\** and *AjCGT1, AjCGT1\**. **g** The relative catalytic activity of the engineered strains of M1–M3. **h** Engineered strains of *E. coli* harboring different combinations (Y1–Y5) of *AjOMT2, AjOMT2^opt, AjOMT2\*^opt* and *AjCGT1, AjCGT1^opt, AjCGT1\*^opt*. **i** The relative catalytic activity of the engineered strains of Y1–Y5; *AjOMT2\**, *AjOMT2-Y203T*; *AjCGT1\**, *AjCGT1-A332F*; *AjOMT2^opt*, codon-optimized *AjOMT2*; *AjOMT2\*^opt*, codon-optimized *AjOMT2-Y203T*; *AjCGT1^opt*, codon-optimized *AjCGT1*; *AjCGT1\*^opt*, codon-optimized *AjCGT1-A332F*; All data represent the means of three parallel experiments and error bars show standard deviation. Source data are provided as a Source data file.

## Strains, plasmids, and media

*E. coli Trans*1-T1 or *E. coli* DH5α was used for the construction and amplification of plasmids. *E. coli* BL21 (DE3) or *E. coli Trans*etta (DE3) was used for protein expression and purification. *E. coli* BL21 (DE3) was used for protein overexpression and production of GA (**2**) and 4-OMGA-Glc (**4**). Plasmids pET28a, pETDuet-1, pCDFDuet-1, and pACYCDuet-1 were purchased from Novagen. Plasmids and strains used in this work were described in Supplementary Data 2 and Supplementary Table 6, respectively.

Luria-Bertani (LB) media containing $10\,g\,L^{-1}$ tryptone, $5\,g\,L^{-1}$ yeast extract, and $10\,g\,L^{-1}$ NaCl were used for plasmid construction, propagation, inoculants preparation, and protein expression. Adjusted M9 media which contain 20 g glucose (M9G), 6.78 g $Na_2HPO_4$, 3 g $KH_2PO_4$, 0.5 g NaCl, 1 g $NH_4Cl$, 0.585 g $MgSO_4$ and 1.1 g $CaCl_2$ per liter were used for biotransformation. Modified M9 (M9Y) media which contain 10 g glycerol, 2.5 g glucose, 6 g $Na_2HPO_4$, 0.5 g NaCl, 3 g $KH_2PO_4$, 1 g $NH_4Cl$, 2 mmol $MgSO_4$, 0.1 mmol $CaCl_2$ and 5 g yeast extract per liter was used for cultivating engineered strains of bioconversion of GA and production of 4-OMGA-Glc.

One liter of fermentation media contained 20 g tryptone, 10 g yeast extract, 3.5 g $K_2HPO_4$, 5.35 g $KH_2PO_4$, 10 g glucose, 0.585 g $MgSO_4$, 4 g $(NH_4)_2SO_4$, 1.7 g citric acid, 1 mL polyether defoamer (CS-4809C), and trace mineral supplement. Trace elements were also

supplemented to all batch cultivations at final concentrations of 7 mg $CuSO_4·5H_2O$, 0.2 mg $CuCl_2$, 0.05 mg $MnSO_4$, 1 mg $FeSO_4·7H_2O$ and 0.225 mg $ZnSO_4·7H_2O$ per liter. To maintain an adequate supply of nutrients during fermentation by supplementing with 150 g glucose, 0.58 g $MgSO_4$, 20 g tryptone, 10 g methionine and 10 g yeast extract per liter. Antibiotics including ampicillin ($50\,\mu g\,mL^{-1}$), kanamycin ($50\,\mu g\,mL^{-1}$), streptomycin ($40\,\mu g\,mL^{-1}$) and chloramphenicol ($34\,\mu g\,mL^{-1}$) were added to the media if needed.

## Analytical procedures

HPLC analysis was performed on an Agilent 1260 HPLC system (Agilent Technologies, Germany) with a Capcell Pak ADME column (250 mm × 4.6 mm I.D., 5 μm, Shiseido Co., Ltd., Japan) at a flow rate of 1 mL/min at 30 °C and injection volume was 10 μL. The mobile phases consisted of methanol or acetonitrile and water with 0.1% formic acid. Linear gradient elution conditions are shown in Supplementary Table 7. 100 μL of samples were mixed by the addition of 200 μL ice-cold methanol, and centrifuged at $15,000\,g$ for 30 min, the supernatant was analyzed by HPLC-MS$^n$ or HPLC. For quantification, three parallel assays were routinely carried out. Enzymatic products were detected on a coupled with an LCQ Fleet ion trap mass spectrometer (Thermo Electron Corp., USA) equipped with an electrospray ionization (ESI) source, and the results were analyzed by Thermo Scientific™ Xcalibur™. Bergenin contents in

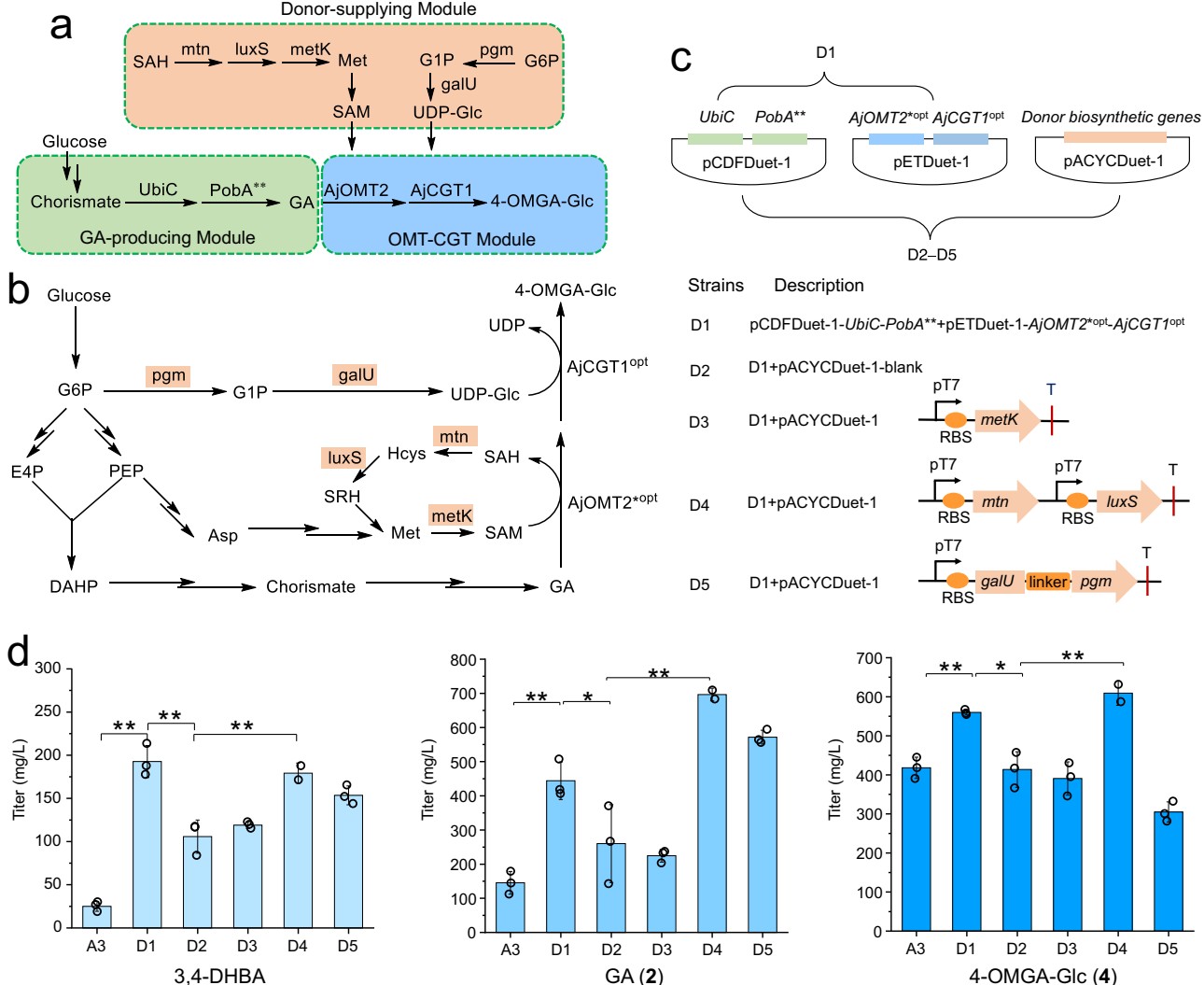

**Fig. 5 | Boosting the SAM and UDP-Glc supplies to improve the yield of 4-OMGA-Glc de novo.** a Modular biosynthetic pathway of 4-OMGA-Glc (**4**) in engineered *E. coli*. b Biosynthetic pathways of UDP-Glc and SAM as well as the SAM regeneration system in *E. coli*. c Reconstruction of engineered strains D1–D5 in *E. coli* BL21. d Evaluation of the engineered strains D1–D5 for titer of 4-OMGA-Glc (**4**) and metabolic intermediates in shake flask fermentation for 48 h. *AjCGT1^opt^*, codon-optimized *AjCGT1*; *AjOMT2^opt^*, codon-optimized *AjOMT2-Y203T* mutation. *S*-adenosyl-*L*-homocysteine (SAH) can be catalyzed by 5′-methylthioadenosine/*S*-adenosylhomocysteine nucleosidase (mtn) to produce *S*-ribosylhomocysteine (SRH), which can be further converted to homocysteine (Hcys) by *S*-ribosylhomocysteine

lyase (luxS). Hcys can be further transformed to methionine (Met), which is finally catalyzed by methionine adenosyltransferase (metK) to form SAM. Phosphoglucomutase (pgm) catalyzes glucose-6-phosphate to form glucose-1-phosphate which is catalyzed by glucose−1-phosphate uridyltransferase (galU) to form UDP-Glc. All data represent the means of three parallel experiments and error bars show standard deviation. Statistical analysis was performed by using the Student's *t* test. *P* value for each comparison from left to right in **d**): 0.0010 (**), 0.0024 (**), 0.0065 (**), 0.0014 (**), 0.0448 (*), 0.0094 (**), 0.0045 (**), 0.0143 (*), 0.0058 (**). Source data are provided as a Source data file.

different organs of *Ardisia japonica* were analyzed as described in Supplementary Method 3. Yields of 3,4-DHBA, GA, 4-OMGA, 4-OMGA-Glc and bergenin were calculated from the peak areas using standard curves of corresponding compounds at 259 nm wavelengths (Supplementary Method 4 and Supplementary Fig. 47).

## Transcriptome sequencing

Fresh materials from 3 different plant parts (leaves, stem, rhizome of *A. japonica*) containing bergenin were selected and sent to Beijing Genomics institution (BGI) for transcriptome sequencing. Transcriptome sequencing and data were acquired on a BGISEQ-500 platform. The unigenes were obtained after assembly and redundancy removal, and were annotated in the following functional databases, NR, NT, SwissProt, KOG, KEGG, GO, and Pfam. The RNA-seq data were utilized to mine biosynthetic genes of bergenin.

## Isolation and cloning of the *C*-glycosyltransferase and *O*-methyltransferase candidate genes from *A. japonica*

The total RNA of *A. japonica* fresh leaves was prepared by using E.Z.N.A.TM Plant RNA Kit (Omega Bio-Tek, USA) and reverse-transcribed (RT) to cDNA with rapid amplification of cDNA ends (RACE) cDNA Amplification Kit (Clontech, USA). The RT product was subjected to RACE according to the manufacturer's protocol. Two CGT candidates (CL7566.Contig1, Unigene14257) were selected by phylogenetic analysis with the known CGTs (Supplementary Fig. 3). The amplified two CGT candidates were designated as AjCGT1 and AjCGT2, respectively. By co-expression analysis with AjCGT1 and AjCGT2, and phylogenetic analysis with the known OMTs (Supplementary Table 4 and Supplementary Fig. 22), two OMT candidate genes (*AjOMT2* and *AjOMT3*) were selected, and the full-length was amplified by PCR.

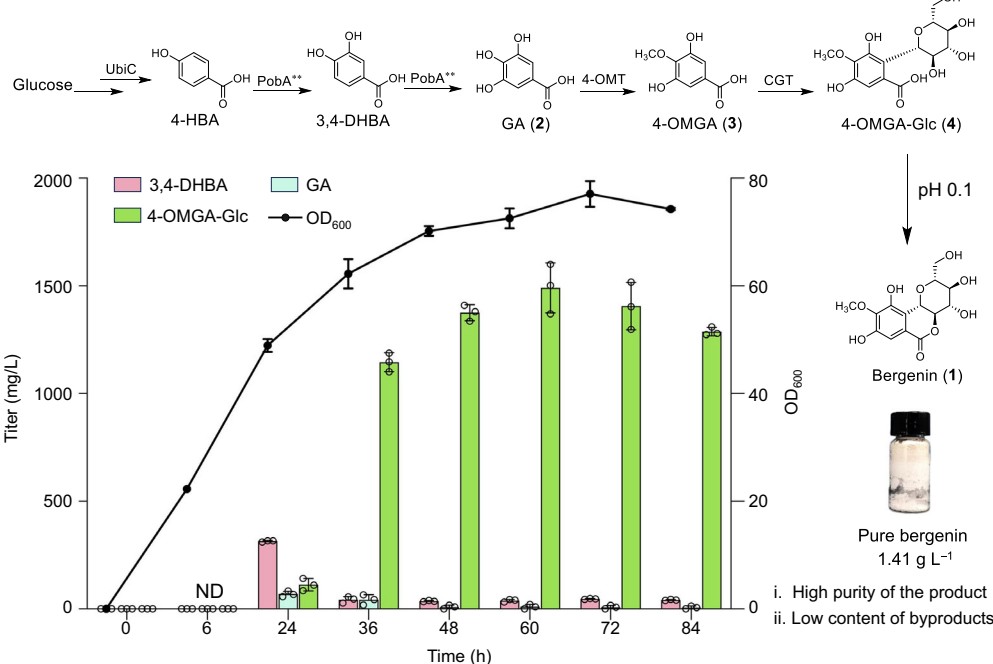

**Fig. 6 | De novo biosynthesis of 4-OMGA-Glc with a 3-L fed-batch bioreactor.** Strain D4 was used for scale-up experiments. Detailed fermentation conditions were shown in "Methods". All the product 4-OMGA-Glc (**4**) was conveniently esterified into bergenin (**1**) by acid treatment and the yield of bergenin reached 1.41 g L$^{-1}$ (Supplementary Fig. 42). ND: not detected. All data represent the means of three parallel experiments and error bars show standard deviation. Source data are provided as a Source data file.

## Expression and purification of recombinant AjCGTs and AjOMTs

The amplified genes were cloned into an *E. coli* expression vector pET28a via BamH I/EcoR I sites using a ClonExpress® One Step Cloning Kit (Vazyme, China), respectively. The recombinant vectors were transformed into *E. coli Trans*etta (DE3) for heterologous expression, respectively. Then the recombinant *E. coli* strain was pre-incubated in 15 mL of LB media containing 50 μg mL$^{-1}$ kanamycin and 34 μg mL$^{-1}$ chloramphenicol and cultured overnight at 37 °C, 10 mL overnight cultures were incubated into 1 L LB-media at 37 °C and 200 rpm until the optical density of the culture at 600 nm (OD$_{600}$) reached 0.6. After addition of isopropyl β-D-thiogalactoside (IPTG) at a final concentration of 0.2 mM, the cells were further cultured at 16 °C for 18 h, then harvested by centrifugation at 10,000 × g for 5 min at 4 °C, and re-suspended in 20 mL binding buffer (20 mM phosphate buffer, 0.5 M NaCl, 20 mM imidazole, pH 7.4). After disruption of the cells by soni-cation, the soluble fraction was passed through a 0.45 μm syringe filter unit, and the supernatant was immediately loaded onto a 1 mL column of Ni-NTA resin (GE, USA) that was pre-equilibrated with binding buffer at 4 °C. The recombinant proteins were then purified with different elution buffer (20 mM phosphate buffer, 0.5 M NaCl, 50–500 mM imidazole, pH 7.4). The target protein was concentrated and buffer was exchanged to desalting buffer (50 mM Tris-HCl buffer, 50 mM NaCl, 1 mM DTT, 5% glycerol, pH 7.4) using a centrifugal concentrator with Amicon Ultra-30K or -10K (Millipore). Protein purity was monitored by SDS-PAGE analysis, and protein concentration for all studies was determined by the Protein Quantitative Kit (*Trans*Gen Biotech, China).

## Enzyme assays of *C*-glycosyltransferases and *O*-methyltransfer-ase in vitro

The in vitro assays of CGTs (in a total volume of 100 μL) consisted of 50 mM Tris-HCl buffer (pH 7.0), 0.8 mM UDP-Glc, 0.4 mM GA or 4-OMGA and 20 μg purified recombinant enzyme, and the mixture was incubated at 40 °C for 1 h. OMT enzyme was assayed in vitro with 1 μL GA or norbergenin (40 mM), 2 μL SAM (40 mM), 2.5 μL MgCl$_2$ (40 mM), 100 μg purified recombinant enzyme, Tris-HCl buffer (pH 7.4, 50 mM)

in a final volume of 100 μL, and the reactions were incubated at 37 °C for 6 h. Control experiments were performed with the boiled enzyme in the same conditions. The reactions were terminated by the addition of 200 μL ice-cold methanol, and the samples were centrifuged at 15,000 g for 30 min and analyzed by HPLC-MS$^n$. The effects of pH, temperature, and metal ions on AjCGT1 and AjOMT2 and the kinetic parameters of these two enzymes were determined as described in Supplementary Methods 5–7.

## Plasmid construction and transformation of biosynthetic pathway

The *AjCGT1* and *AjOMT2* genes were inserted into the expression plasmid pETDuet-1, pCDFDuet-1, and pACYCDuet-1, respectively. *AjOMT2* was introduced into the *EcoR*I and *Not*I sites, *AjCGT1* was introduced into the *EcoR*V and *Xho*I sites of the plasmid to generate pETDuet-*AjOMT2-AjCGT1*, pCDFDuet-*AjOMT2-AjCGT1*, pACYCDuet-*AjOMT2-AjCGT1*. The three plasmids were transferred to *E. coli* BL21 (DE3) to obtain engineered strains E1–E3, respectively (Supplementary Table 6). Other plasmids used in the biosynthesis of bergenin were constructed in the same way, the obtained plasmids were used for transformation to *E. coli* BL21 (DE3).

## Biotransformation of GA by engineered strains

Single colonies of strains E1–E3 were incubated in 3 mL of LB media containing antibiotic and cultured overnight at 37 °C, then 100 μL overnight cultures were inoculated into 10 mL LB media containing antibiotic at 37 °C and 200 rpm until OD$_{600}$ reached 0.6, respectively. Protein expression was then induced by adding 0.2 mM IPTG at 16 °C for 18 h. After induction, *E. coli-AjOMT2-AjCGT1* cells were harvested by centrifugation and washed with M9G media. The engineered strain was then resuspended in M9G media at 30 °C and the cell density was adjusted to an OD$_{600}$ of 6.0. GA was added to the culture media at the final concentration of 0.4 mM, after further incubation for 24 h, the culture solution was analyzed by HPLC. The biotransformation experiments for optimized engineered strains were consistent with the

above method except that 100 μL overnight cultures of engineered strains were inoculated directly into M9Y media to simplify the operation procedures and the final concentration of GA was adjusted to 0.8 mM in the feeding experiments of codon-optimized strains Y1–Y5.

### Construction of GA-producing strains
Codon-optimized *pobA-Y385F/T294A* was synthesized by Generay Biotechnology Co., Ltd., and *UbiC* was amplified from the genome of *E. coli*. Two genes were introduced into the expression plasmid pETDuet-1, pCDFDuet-1, and pACYCDuet-1, respectively. Then three plasmids were transferred to *E. coli* BL21 (DE3) to obtain engineered strains S1–S3, respectively (Supplementary Table 6). The extended culture of engineered strains and protein-induced expression were performed as operated as described above, respectively. The harvested cells were then resuspended in 20 mL M9G media and the cell density was adjusted to an $OD_{600}$ of 6.0. Each supernatant of the fermented engineered strains was analyzed by HPLC after 24 h of culture.

### De novo biosynthesis of 4-OMGA-Glc
Strains S1–S3 were transformed with pETDuet-*AjOMT2-AjCGT1*, pCDFDuet-*AjOMT2-AjCGT1,* and pACYCDuet-*AjOMT2-AjCGT1*, respectively, generating the recombinant strains A1–A6. Transformants were pre-inoculated in 10 mL LB overnight and then inoculated 100 μL into 10 mL M9Y media containing suitable antibiotics. The cultures were cultivated at 37 °C until $OD_{600}$ reached 0.8 and then induced with IPTG at the final concentration 0.5 mM at 16 °C. After 18 h of induction, the cultivation temperature was adjusted to 30 °C. After another 48 h, samples were collected and the product 4-OMGA-Glc as well as its intermediates were analyzed by HPLC. Other engineered strains about optimization of biosynthetic pathway were conducted using the same methods (Supplementary Table 6).

### Homology modeling, molecular docking, and site-directed mutagenesis of AjOMT2 and AjCGT1
The SWISS-MODEL server (https://swissmodel.expasy.org/) was used to produce a three-dimensional structure of the protein. Structural models of AjOMT2 were generated according to the reported crystal structure of sorghum caffeoyl-CoA *O*-methyltransferase (CCoAOMT, PDB code 5KVA)[46], which had the highest sequence identity (55.7%). The Ramachandran plot of AjOMT2 indicated that 91.9% of all residues were in favored regions and 100% of all residues were in allowed regions (Supplementary Fig. 48). Structural models of AjCGT1 were generated according to the reported crystal structure of SbCGTa (PDB code 6LG0) in complex with UDP-Glc[30], which had the highest sequence identity (49.8%). The Ramachandran plot of AjCGT1 indicated that 92.7% of all residues were in favored regions and that 99.7% of all residues were in allowed regions (Supplementary Fig. 49). Based on these statistics, the quality of the modeled protein can be ensured.

Molecular docking between AjOMT2 and GA (**2**) was performed with AutoDock Vina. Ligands were prepared by using AutoDock Tools to add charges and hydrogen atoms. The probable binding sites of SAM and the substrate were aligned from the template. The grid box was 80 × 80 × 80 Å with a grid point spacing of 0.5 Å. The docking results were visualized and analyzed by PyMOL. The same method was used for molecular docking between AjOMT2 and GA-Glc (**5**). Molecular docking between AjCGT1 and UDP-Glc was also performed in AutoDock Vina, and AjCGT1 in complex with UDP-Glc was molecularly docked with GA (**2**) and 4-OMGA (**3**), respectively.

Site-directed mutagenesis by PCR amplification of pETDuet-*AjOMT2-AjCGT1* was performed using KOD One PCR Master Mix and digestion with DMT enzyme (Dpn I) in DMT competent cells (TransGen Biotech, China). Sequence confirmation of the pETDuet-*AjOMT2-AjCGT1* mutants was performed, and the resulting plasmids were subsequently transformed into *E. coli* BL21 (DE3) for biotransformation experiments. The mutation-generating primer pairs used are listed in Supplementary Data 1.

### Fed-batch fermentation
Strain D4 was activated on solid media supplemented with antibiotics (Amp$^r$, Str$^r$, and Chl$^r$), and well-stacked colonies were selected and inoculated in 5 mL of LB media supplemented with three antibiotics at 37 °C and 200 rpm for 12 h to produce primary seed cultures. Then, 400 μL of each seed culture was transferred to 40 mL of fresh LB medium and incubated at 37 °C and 200 rpm for 12 h to yield the second seed culture. All second seed media were inoculated with 1 L of modified fermentation media in a bioreactor with a maximal working volume of 3 L and cultivated at 37 °C. When the $OD_{600}$ reached approximately 20, 0.5 mM IPTG was added to induce protein expression, the temperature was increased to 16 °C, and 10 mL of the feeding solution was added per hour. After 17 h of induction, the temperature was adjusted to 30 °C, the pH was maintained at 7.0, and the dissolved oxygen concentration (DOC) was maintained at 30% throughout the fermentation process. Samples were taken at regular intervals to measure the $OD_{600}$, and the supernatants were subjected to HPLC analysis. Extraction, isolation and purification of bergenin (**1**) from cultures of bioreactor were performed as described in Supplementary Method 8.

### Reporting summary
Further information on research design is available in the Nature Portfolio Reporting Summary linked to this article.

## Data availability
The gene sequences of *AjCGT1*, *AjCGT2*, *AjOMT2,* and *AjOMT3* are deposited in NCBI under accession OR672142, OR672143, OR672144, and OR672145, respectively. Source data are provided with this paper.

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

## Acknowledgements

This work is financially supported by the National Key Research and Development Program of China (No. 2023YFA0914100 to J.D.), the National Natural Science Foundation of China (No. 82373995 to L.Y.) and CAMS Innovation Fund for Medical Sciences (No. 2021-I2M-1-029 to K.X.).

## Author contributions

J.D., L.Y. and K.X. conceived the study; R.Y., B.X. and Q.L. performed the experiments and analyzed the data; K.X., S.S., S.W., D.C., J.L. and R.C. assisted with experimental performance; K.X., J.D. and L.Y. analyzed the data and wrote the manuscript; All authors read and confirmed the manuscript.

## Competing interests

The authors declare no competing interests.
