## [Peer Review File · Nature Communications]

Unravelling and optimizing the biosynthetic pathway of bergeninReviewers' Comments:

Reviewer #1:

Remarks to the Author:

In this manuscript, the authors isolated and characterized UGT and MT those would be related to the biosynthesis of bergenin from *Ardisia japonica*. Then the authors attempted to reconstruct the biosynthetic pathway of bergenin in *E. coli*, and finally achieved de novo production of bergenin precursors equivalent to 1.41 g/L of bergenin in a bioreactor through a several-step metabolic engineering approach.

As the authors mentioned in the text, bergenin is an important natural medicine used for bronchitis and tuberculosis, and has recently attracted attention for its various bioactive properties, but it is rare in nature and is found only low amounts in limited plant species. Therefore, this study is an important achievement leading to the microbial production of bergenin.

I suggest some improvements and corrections of this manuscript as follows.

1. Evaluation of the enzyme activity

The enzymatic parameter shown in this paper suggested that the activity of AjCGT1 toward GA ($k_{cat}/K_m = 320 \text{ M}^{-1} \text{ s}^{-1}$ which calculated for the presented data) is quite low comparing to the reported other CGTs belongs to UGT708 (for examples, miCGT showed $k_{cat}/K_m = 34000 \text{ M}^{-1} \text{ s}^{-1}$ for maclurin, FeCGTa showed $k_{cat}/K_m = 850000 \text{ M}^{-1} \text{ s}^{-1}$ for 2-hydroxynaringenin). It required a significant amount of enzyme (10–20 μg) for each reaction. This could suggest that GA is not the primary substrate for AjCGT1. It may function as a biosynthetic enzyme even if its activity is low, but please discuss that as well.

The authors emphasize that the reactivity of AjCGT1 is "different from other reported plant CGTs" (l. 400, Summery), but its reactivity with common CGT substrates having trihydroxyacetophenone-like structure (such as phloretin, 2-hydroxynaringenin or maclurin) has not been evaluated. It is possible that this enzyme is more reactive with these substrates. I think substrate preference of AjGT1 would need to be confirmed for this discussion.

Also, the activity of AjMT2 also appears to be weaker than other MTs reported. Please discuss this as well.

2. This paper contains many expressions of "first time" (l. 25, l. 29, l. 85, l. 131, l.196, l. 393, l.398, l. 408). But it is up to the readers to judge whether it is the first time or not, so I would suggest reducing this description.

Certainly, the enzymes exhibiting this activity have not been reported to date, but candidate biosynthetic gene sequences such as CGTs and MTs from *Ardisia japonica* and related species (*Bergenia purpurascens*) have recently been opened to the database by another research group (Genbank Accession Nos. OR267195–98). Therefore, it is not certain that if this report is the 'first' identification and characterization of CGTs showing this activity.

3. Fed-batch production of 4-OMGA-Glc in bioreactor (l. 378–392)

The authors mentioned that "GA titer in such fed-batch was nearly all consumed by the downstream enzyme" (l.381) and "low content of byproduct" (l. 388), but we cannot check them directly in Fig. 6. To clarify the status, please show the HPLC chart of the result of the production of 4-OMGA-Glc in bioreactor as supplementary figures.

Also, please show the HPLC charts before and after adjusting pH involved in the conversion of 4-OMGA-Glc into bergenin (the reaction corresponding to the Supplementary Fig. 21) so that the readers can confirm the condition of the conversion.

It also needs to explain how to purify the product before NMR in the method part pf supplementary information.

4. Mediums used for the bioconversion.

The authors used the term "M9 medium" as the M9 medium containing 2 % glucose. However, it is

confusing because we usually think of M9 medium as a separate carbon source or about 0.4% Glc. If you want to abbreviate it, it would be better to define and use another term, for example, M9G or M9Glc, as well as M9Y.

In addition, why did the authors change the medium from M9 containing 2% Glc to M9Y? Please explain briefly in the text.

5. UGT nomenclature

I recommend consulting UGT nomenclature committee to give AjCGT1 a UGT name so that it is easy to understand what kind of UGT it is.

6. Other minor concerns

1) l. 143–147 "Next, engineering strain harboring AjCGT1 gene was constructed and successfully converted exogenously added 2 and 3 to 5 and 4, respectively, indicating..."

No data on this topic have been presented. Please show results as supplementary figures.

2) Fig 2b and supplementary Table 1 Bergenin content (%)

Please clarify what you mean by percentages. Per dry weight?

3) Fig.4a and Supplementary Fig. 19

AjCGT1-GA and AjCGT1-4OMGA

The two proteins have the same name but differ in an amino acid residue (332). Please change the name for clarify.

Structure of 4OMGA

It appears to be a methyl group, not a methoxy group. Please confirm and correct it.

4) l. 215 "after adding exogenous 2 through whole-cell biocatalysis"

Please specify the substrate concentration not only in the method, but also here and in the legend for Supplementary Fig. 11.

5) l. 292–296 This sentence is difficult to understand. What is the hot spot? No data or reference have been shown. Please make clear.

6) Typos and Terms

l. 450 "20 tryptone" would be "20 g tryptone"

l. 452 What is "foam suppressor" ? Please specify.

l. 459 "chloramphenicol" and l. 503 "chloromycetin"

Please use the same term for the same compound. I think chloramphenicol is more common.

Supplementary Fig. 1 "GuCGT" would be "CuCGT". Please confirm it.

Reviewer #2:

Remarks to the Author:

Yan et al. elucidated two key steps in the biosynthetic pathway of bergenin. AjCGT and AjOMT catalyze 2-C-glycosylation and O-methylation, respectively. Furthermore, they reconstructed the biosynthetic pathway in *Escherichia coli* by co-expressing AjCGT, AjOMT and upstream genes, and achieved de novo biosynthesis of 4-OMGA-Glc through plasmid expression. Bergenin could be obtained after acid treatment. Finally, the production of bergenin could reach 1.41 g/L in a 3-L bioreactor. This work provides a complete story and demonstrates the powder of synthetic biology in preparing complex natural products. However, studies on biochemistry, structure and function of key enzyme are relatively weak. The catalytic mechanisms of 2CGT and OMT should be investigated to improve novelty and quality of this work. Otherwise, the reviewer would not recommend this work for publication in *Nat. Commun.*

Major concerns:

1. What's the major difference between AjCGT and previously reported CGTs? While AjCGT shares highly similarity with UGT708 family glycosyltransferases (as shown in Fig. S1), the authors claim that 'Different C-glycosylation property from other reported plant CGTs, the two CGTs displayed the ability to introduce C-sugar into the aromatic ring with acyl 3,4,5-pyrogallol structure, suggesting that they might be a novel type of plant CGT'. However, the evidence provided is limited. Whether other CGTs could catalyze 2-C-glycosylation of 2 or 3 needs to be investigated.
2. It would be valuable to investigate the relationship between the sequence and structure of AjCGT. Crystal structure analysis and theoretical calculation would provide important approaches to dissect the mechanisms.

Minor concerns:

1. The low affinity ($K_m=279.6 \mu\text{M}$) of AjCGT1 towards 3 is unexpected. It seems to indicate that 3 is not the endogenous substrate for AjCGT1. Have the authors used 2 as sugar acceptor to measure the kinetic parameter of AjCGT1?
2. Line 173, the authors mentioned that the relative rate of AjOMT2 catalyzing 6 to generate 1 is low. The reviewer did not find obvious difference between Fig 2k and Fig 2n.
3. The catalytic efficiencies of AjCGT2 and AjOMT3 towards different acceptors should be measured, respectively.
4. There seem to be some mistakes in the ligands labeled in Figures S4, S5, S8, and S9. The authors should label which buffer was used for the enzyme property studies in the figures.
5. Figure S4, AjCGT1 shows a relative conversion rate of approximately 15% and 80% in Buffer A (pH=7) and Buffer B (pH=7), respectively. Please confirm it.

Reviewer #3:

Remarks to the Author:

The manuscript by Yan et al. elucidated the biosynthetic pathway of bergenin in *Ardisia japonica* and constructed the de novo biosynthetic pathway of the direct precursor of bergenin. Although this work established a foundation for biosynthesis of bergenin, the novelty of this research is not enough and the writing also is careless.

1. The distribution of bergenin in medicinal plants is very wide, but no reference was provided. It is necessary to discuss the possibility of the synthetic pathway of bergenin from other species.
2. It is important to screen the function of other OMT genes which co-expressed with the bait genes GTs.
3. "the relatively higher conversion rates of AjCGT1 than those of AjCGT2 with GA (2) and 4-OMGA (3)". But in Fig2, there was no significant difference.
4. In Supplementary Fig. 4, 5, 8 and 9, (a) and (b) of these graphs do not correspond to the headings.
5. When added the codon-optimized strategy, the strain Y4 harboring codon-optimized AjOMT2*opt and AjCGT1opt (codon-optimized strain M1) showed the highest catalytic activity, not codon-optimized strain M2 (carrying pET-AjOMT2-AjCGT1*).
6. There is wrong description in Fig. 4h.
7. The obtained yields of bergenin from *E. coli* exhibit promising results; however, it is essential to compare it to the yields achieved through analogous engineering methodologies for related compounds.

Response Section

Contents

Response to Reviewer #1	Page 2–23
Response to Reviewer #2	Page 24–46
Response to Reviewer #3	Page 47–54

Response to Reviewer #1:

Comments:

In this manuscript, the authors isolated and characterized UGT and MT those would be related to the biosynthesis of bergenin from *Ardisia japonica*. Then the authors attempted to reconstruct the biosynthetic pathway of bergenin in *E. coli*, and finally achieved *de novo* production of bergenin precursors equivalent to 1.41 g/L of bergenin in a bioreactor through a several-step metabolic engineering approach.

As the authors mentioned in the text, bergenin is an important natural medicine used for bronchitis and tuberculosis, and has recently attracted attention for its various bioactive properties, but it is rare in nature and is found only low amounts in limited plant species. Therefore, this study is an important achievement leading to the microbial production of bergenin.

I suggest some improvements and corrections of this manuscript as follows.

Response #1:

We would like to thank you very much for your time to review our manuscript. Your expertise suggestions and invaluable comments to our work are very helpful to improve the quality of our work and manuscript. We have addressed each point below.

1. Evaluation of the enzyme activity

The enzymatic parameter shown in this paper suggested that the activity of AjCGT1 toward GA ($k_{cat}/K_m = 320 \text{ M}^{-1} \text{ s}^{-1}$ which calculated for the presented data) is quite low comparing to the reported other CGTs belongs to UGT708 (for examples, miCGT showed $k_{cat}/K_m = 34000 \text{ M}^{-1} \text{ s}^{-1}$ for maclurin, FeCGTa showed $k_{cat}/K_m = 850000 \text{ M}^{-1} \text{ s}^{-1}$ for 2-hydroxynaringenin). It required a significant amount of enzyme (10–20 μg) for each reaction. This could suggest that GA is not the primary substrate for AjCGT1. It may function as a biosynthetic enzyme even if its activity is low, but please discuss that as well.

The authors emphasize that the reactivity of AjCGT1 is "different from other reported plant CGTs" (l. 400, Summery), but its reactivity with common CGT

substrates having trihydroxyacetophenone-like structure (such as phloretin, 2-hydroxynaringenin or maclurin) has not been evaluated. It is possible that this enzyme is more reactive with these substrates. I think substrate preference of AjGT1 would need to be confirmed for this discussion.

Also, the activity of AjMT2 also appears to be weaker than other MTs reported. Please discuss this as well.

Response #1-1:

Thank you so much for your professional advice and questions. Yes, the apparent enzymatic parameter indeed suggested that the activity of AjCGT1 toward GA is quite low. As mentioned by the reviewer, this suggested that GA may not be the primary substrate for AjCGT1 although it may function as a biosynthetic enzyme. We agree with this point and according to the suggestions, we have added description in the manuscript.

Actually, when we obtained the enzymatic parameters (such as $k_{cat}/K_m = 320 \text{ M}^{-1} \text{ s}^{-1}$ calculated by the reviewer) of AjCGT1, we re-performed the assays and re-checked the data for several times. We found that the GA (**2**) and 4-OMGA (**4**) are unstable in the reaction buffers. After checking the literatures, it was reported that gallic acid compounds are not only very unstable under neutral and alkaline conditions, but also easily undergoes non-specific binding with proteins due to its pyrogallol structure.^{1,2} According to the results of our stability experiments, GA (**2**) decreased spontaneously and gradually in neutral and alkaline buffers such as Tris-HCl and phosphate buffers, and it was even completely disappeared within 10 minutes in phosphate buffer (**Figure r1-1** and **Figure r1-2**). In addition, the 4-OMGA (**3**) was also slightly decreased in buffer solutions (**Figure r1-3**). These figures were also added to the **Supplementary Material** (new version) as **Supplementary Fig. 5–7**. Thus, the apparent kinetic parameters of AjCGT1 we obtained here may not be its real kinetic parameters and can't reflect its enzymatic property including the catalytic efficiency.

Figure r1-1. Investigation of the stability of GA (2) in different buffers. GA (2) was added into MeOH (control) and the corresponding buffers for 10 minutes to evaluate its stability. HPLC analysis results showed that the peak areas of GA (2) in MeOH, pH7.0 Tris-HCl, pH8.0 Tris-HCl, pH7.0 $\text{Na}_2\text{HPO}_4\text{-NaH}_2\text{PO}_4$ and pH8.0 $\text{Na}_2\text{HPO}_4\text{-NaH}_2\text{PO}_4$ buffer were 548 (control), 513, 412, 495, and 0, respectively. These results indicated GA (2) could be spontaneously decreased even in a short reaction time (10 min).

Figure r1-2. Investigation of the stability of GA (2) at different incubation times. GA (2) was added into pH7.0 $\text{Na}_2\text{HPO}_4\text{-NaH}_2\text{PO}_4$ buffer and was incubated for 10, 20, 30, and 45 min, respectively. HPLC analysis results showed that the peak areas of GA (2) are 548 (control), 495 (10 min), 429 (20 min), 408 (30min) and 315 (45 min), respectively. These results indicated that GA (2) was unstable in phosphate buffer solution.

Figure r1-3. Investigation of the stability of 4-OMGA (3) at different incubation times. 4-OMGA (3) was added into pH 8.0 Na₂HPO₄-NaH₂PO₄ buffer and was incubated for 0, 10, 20, 30, and 60 min, respectively. HPLC analysis results showed that the peak areas of 4-OMGA (3) are 1284 (control), 1174 (0 min), 1149 (10 min), 1129 (20 min), 1095 (30min) and 954 (60 min), respectively. These results indicated that 4-OMGA (3) was unstable in phosphate buffer solution.

As for the catalytic activity of AjCGT1 to the other substrates, we evaluated the catalytic activity of AjCGT1 to the substrates having 2, 4, 6-trihydroxyacetophenone-like structure including phloretin, 2-hydroxynaringenin, maclurin, apigenin and 5, 7-dihydroxycoumarin (**Scheme r1-1**), which are substrates of common reported CGTs.

Scheme r1-1. Probing the catalytic activity of AjCGT1 with substrates having 2, 4, 6-trihydroxyacetophenone-like structure

AjCGT1 showed C-glycosylating activity to all these five representative substrates (**Figures r1-4–r1-8**). Given the apparent K_m value of AjCGT1 to GA (**2**) cannot reflect the real affinity, we further investigated the types of C-glycosides in *Ardisia japonica*. Although AjCGT1 showed C-glycosylating activity to these five substrates, their corresponding C-glycosylated products have not been isolated from *A. japonica*. So far, only C-glycosides having 3,4,5-trihydroxybenzoic acid-like structure (bergenin and its derivatives) were isolated from *A. japonica*. We speculated the C-glycosyltransferase AjCGT1 was responsible for the formation of bergenin and its derivatives. As suggested, we have discussed this in the manuscript and added the **Figures r1-4–r1-8** to the **Supplementary Material** (new version) as **Supplementary Fig. 10–14**.

Figure r1-4. HPLC-MS/MS² analysis of AjCGT1 catalyzing the C-glycosylation of phloretin. a) AjCGT1 catalyzed the C-glycosylation of phloretin; b) HPLC chromatograms of the AjCGT1 catalyzing reaction, control and the standard C-glycosides; c) Typical negative ion MS and MS² spectra for the C-glycosylated products **p1** and **p2**.

Figure r1-5. HPLC-MS/MS² analysis of AjCGT1 catalyzing the C-glycosylation of 2-hydroxynaringenin. a) AjCGT1 catalyzed the C-glycosylation of 2-hydroxynaringenin; b) HPLC chromatograms of the AjCGT1 catalyzing reaction, control and the standard C-glycosides; c) Typical positive ion MS and MS² spectra for the C-glycosylated products p3/p4.

Figure r1-6. HPLC-MS/MS² analysis of AjCGT1 catalyzing the C-glycosylation of maclurin. a) AjCGT1 catalyzed the C-glycosylation of maclurin; b) HPLC chromatograms of the AjCGT1 catalyzing reaction, control and the standard C-glycosides; c) Typical negative ion MS and MS² spectra for the C-glycosylated product **p5**.

Figure r1-7. HPLC-MS/MS² analysis of AjCGT1 catalyzing the C-glycosylation of apigenin. a) AjCGT1 catalyzed the C-glycosylation of apigenin; b) HPLC chromatograms of the AjCGT1 catalyzing reaction and the control; c) Typical negative ion MS and MS² spectra for the C-glycosylated product **p6** and O-glycosylated product **p7**.

Figure r1-8. HPLC-MS/MS² analysis of AjCGT1 catalyzing the C-glycosylation of 5,7-dihydroxycoumarin. a) AjCGT1 catalyzed the C-glycosylation of 5,7-dihydroxycoumarin; b) HPLC chromatograms of the AjCGT1 catalyzing reaction, control and the standard C-glycoside; c) Typical negative ion MS and MS² spectra for the C-glycosylated product **p8**.

2. This paper contains many expressions of "first time" (l. 25, l. 29, l. 85, l. 131, l.196, l. 393, l.398, l. 408). But it is up to the readers to judge whether it is the first time or not, so I would suggest reducing this description.

Certainly, the enzymes exhibiting this activity have not been reported to date, but candidate biosynthetic gene sequences such as CGTs and MTs from *Ardisia japonica* and related species (*Bergenia purpurascens*) have recently been opened to the database by another research group (Genbank Accession Nos. OR267195–98). Therefore, it is not certain that if this report is the 'first' identification and characterization of CGTs showing this activity.

Response #1-2:

Thanks a lot. We agree with your expertise comments and suggestions, and we have deleted the expression of "first time" in the manuscript.

Yes, four candidate biosynthetic genes (Accession Nos. OR267195–98) of CGTs and MTs from *Bergenia purpurascens* and *Ardisia japonica* have recently been opened to the NCBI database. We aligned the sequences of AjCGT1 and AjCGT2 with those of the candidate CGTs (OR267197 and OR267198), respectively. AjCGT1 and AjCGT2 showed about 90% identity to OR267197 and about 50% identity to OR267198, respectively (**Table r1-1**).

Table r1-1 Protein identities of AjCGT1, AjCGT2, OR267197 and OR267198

Protein Identity	OR267197	OR267198
AjCGT1	90.2%	50.9%
AjCGT2	90.5%	51.6%

The sequences of AjOMT2, AjOMT3, OR267195 and OR267196 were also analyzed. AjOMT2 and AjOMT3 showed about 10% identities to OR267195 and OR267196, respectively (**Table r1-2**).

Table r1-2 Protein identities of AjOMT2, AjOMT3, OR267195 and OR267196

Protein Identity	OR267195	OR267196
AjOMT2	11.7%	9.6%
AjOMT3	10.9%	9.4%

According to the sequence identities, especially for that of AjCGT1, AjCGT2 and OR267197, these CGTs may have similar C-glycosylation activities. **However, no identification and characterization work of these candidate enzymes (OR267195–98) has been reported before we submitted this work to *Nature Communications*.** Thanks for the helpful suggestions and we have deleted the description of "first time".

3. Fed-batch production of 4-OMGA-Glc in bioreactor (l. 378–392)

The authors mentioned that "GA titer in such fed-batch was nearly all consumed by the downstream enzyme" (l.381) and "low content of byproduct" (l. 388), but we cannot check them directly in Fig. 6.

To clarify the status, please show the HPLC chart of the result of the production of 4-OMGA-Glc in bioreactor as supplementary figures.

Also, please show the HPLC charts before and after adjusting pH involved in the conversion of 4-OMGA-Glc into bergenin (the reaction corresponding to the Supplementary Fig. 21) so that the readers can confirm the condition of the conversion.

It also needs to explain how to purify the product before NMR in the method part of supplementary information.

Response #1-3:

Many thanks for such helpful comments and suggestions. The HPLC chromatograms of the production of 4-OMGA-Glc (**4**) in bioreactor at different time periods were provided as **Figure r1-9 (Supplementary Fig. 44** in the new version) to clarify the status.

Figure r1-9. HPLC analysis of the production of 4-OMGA-Glc (4) in bioreactor at different time periods. a) The artificial biosynthetic pathway to 4-OMGA-Glc from glucose in engineered strain D4; b) HPLC analysis of the fermentation broth supernatants of strain D4 in bioreactor and shake flask at different time periods, respectively. (**Supplementary Fig. 44** in the new version)

According to your suggestions, the HPLC charts before and after adjusting pH involved in the conversion of 4-OMGA-Glc (4) into bergenin (1) were supplemented in **Figure r1-10** and also provided as **Supplementary Fig. 45** in the **Supplementary Material**.

Figure r1-10. The conversion of 4-OMGA-Glc (4) into bergenin (1) by pH adjustment of the cultures. a) The reaction of 4-OMGA-Glc (4) in the acid environment; b) The conversion rates of 4-OMGA-Glc (4) at different pH values; b) HPLC analysis of the conversion of 4-OMGA-Glc (4) in the cultures at different pH values. (**Supplementary Fig. 45** in the new version)

In addition, the detailed process of isolation and purification of bergenin (1)

were added in the experimental section of **Supplementary Material**.

4. Mediums used for the bioconversion.

The authors used the term "M9 medium" as the M9 medium containing 2 % glucose. However, it is confusing because we usually think of M9 medium as a separate carbon source or about 0.4% Glc. If you want to abbreviate it, it would be better to define and use another term, for example, M9G or M9Glc, as well as M9Y.

In addition, why did the authors change the medium from M9 containing 2% Glc to M9Y? Please explain briefly in the text.

Response #1-4:

Thanks a lot for your kind reminding and the question. The term "M9G" was used to designate M9 medium containing 2 % glucose in the text.

The reason for changing the medium from M9 containing 2% Glc to M9Y is to simplify the steps of the biosynthesis process. At first, M9G medium was used. Engineered strain was firstly inoculated into LB medium, after induction, engineered strain cells were harvested by centrifugation and washed by M9G medium. Then engineered strain was resuspended in M9G medium for further cultivation. When M9G was changed to M9Y, there is no need for centrifugation and medium replacement. Therefore, the aim of changing the medium from M9G to M9Y was to simplify the operation procedures. According to the suggestions, the reason for changing the culture medium was explained in the main text.

Line603 (new version): "...To simplify the operation procedures, M9G was replaced by M9Y medium..."

5. UGT nomenclature

I recommend consulting UGT nomenclature committee to give AjCGT1 a UGT name so that it is easy to understand what kind of UGT it is.

Response #1-5:

Thank you very much for your expertise. According to the UGT nomenclature committee, AjCGT1 and AjCGT2 were named as UGT708AL1 and UGT708AL2, respectively. And these two UGT names were provided in the main text.

6. Other minor concerns

1) I. 143–147 "Next, engineering strain harboring *AjCGT1* gene was constructed and successfully converted exogenously added **2** and **3** to **5** and **4**, respectively, indicating..."

No data on this topic have been presented. Please show results as supplementary figures.

Response #1-6-1:

Thanks for your kind reminding. The corresponding HPLC charts were provided (**Figures r1-11** and **r1-12**) and added to the supplementary figures (**Supplementary Fig. 20** and **21** in the new version).

Figure r1-11. The transformation of GA (2) into GA-Glc (5) by *E. coli-AjCGT1*. a) *E. coli-AjCGT1* catalyzed the C-glycosylation of GA (2). b) HPLC analysis of the whole cell catalytic product of *E. coli-AjCGT1* and the reaction catalyzed by AjCGT1 *in vitro*. c) Typical negative ion MS and MS² spectra for the product 5. (**Supplementary Fig. 20** in the new version)

Figure r1-12. The transformation of 4-OMGA (3) into 4-OMGA-Glc (4) by *E. coli-AjCGT1*. a) *E. coli-AjCGT1* catalyzed the C-glycosylation of 4-OMGA (3). b) HPLC analysis of the whole cell catalytic product of *E. coli-AjCGT1*, the control reaction and the standard. c) Typical negative ion MS and MS² spectra for the product 4. (**Supplementary Fig. 21** in the new version)

2) Fig 2b and supplementary Table 1 Bergenin content (%)

Please clarify what you mean by percentages. Per dry weight?

Response #1-6-2:

Many thanks for your expertise. Yes, it is per dry weight and dry weight was

added to the bergenin content (%) in Fig 2b and Supplementary Table 1.

3) Fig.4a and Supplementary Fig. 19

AjCGT1-GA and AjCGT1-4OMGA

The two proteins have the same name but differ in an amino acid residue (332).

Please change the name for clarify.

Structure of 4OMGA

It appears to be a methyl group, not a methoxy group. Please confirm and correct it.

Response #1-6-3:

Thanks for your professional comments and advices. We have carefully checked the protein structures and revised the labels of the amino acid residues. The methoxy group of the substrate has also been corrected. In addition, the catalytic mechanism of AjCGT1 was explored as suggested by the other reviewer. So, homology modelling and molecular docking was re-performed, and some new potential key sites were obtained and also labelled in **Fig 4a** and **Supplementary Fig 15** in the new version of **Supplementary Material**.

4) I. 215 "after adding exogenous **2** through whole-cell biocatalysis"

Please specify the substrate concentration not only in the method, but also here and in the legend for Supplementary Fig. 11.

Response #1-6-4:

Many thanks for this great suggestion. Accordingly, the concentration of exogenous **2** ("...with the final concentration of 0.4 mM...") was added in the manuscript and in the legend for **Supplementary Fig. 11 (Supplementary Fig 34** in the new version).

5) I. 292–296 This sentence is difficult to understand. What is the hot spot? No data or reference have been shown. Please make clear.

Response #1-6-5:

Thank you for your nice suggestion. Hot spot Wizard is web server for automated design of mutations and smart libraries based on sequence input information. According to the suggestion, we have added detailed information about the hot-spot (“...HotSpot Wizard is a web server focused on automated prediction of hot-spot residues for mutagenesis that are likely to alter enzyme activity...”) and the reference was also provided in the main text (ref 47 Sumbalova, L., et al. HotSpot Wizard 3.0: web server for automated design of mutations and smart libraries based on sequence input information. *Nucleic Acids. Res.* **46**, W356–W362 (2018)).

6) Typos and Terms

I. 450 "20 tryptone" would be "20 g tryptone"

I. 452 What is "foam suppressor" ? Please specify.

I. 459 "chloramphenicol" and I. 503 "chloromycetin"

Please use the same term for the same compound. I think chloramphenicol is more common.

Supplementary Fig. 1 "GuCGT" would be "CuCGT". Please confirm it.

Response #1-6-6:

Thank you very much for your carefulness. Accordingly, we have carefully checked and revised the Typos and Terms.

I. 450 "20 tryptone" has been modified to "20 g tryptone".

This “foam suppressor” is a kind of polyether defoamer (CS-4809C) that is used to control or reduce the foam producing during fermentation. To make the description clear, we have revised “foam suppressor” to “polyether defoamer (CS-4809C)”. Detailed information is also added in the part of **Plant materials and chemical reagents**.

I. 503 "chloromycetin" has been replaced by "chloramphenicol".

Supplementary Fig. 1 "GuCGT" has been corrected to "CuCGT".

References

1. Zhang, J., et al. Changes in composition and enamel demineralization inhibition activities of gallic acid at different pH values. *Acta Odontol. Scand.* **73**, 595–601 (2015).
2. Nascimento, J. M. D., et al. Evaluation of the influence of temperature on the protein-tannic acid complex. *Int. J. Biol. Macromol.* **182**, 2056–2065 (2021).

Response to Reviewer #2

Comments:

Yan et al. elucidated two key steps in the biosynthetic pathway of bergenin. AjCGT and AjOMT catalyze 2-C-glycosylation and O-methylation, respectively. Furthermore, they reconstructed the biosynthetic pathway in *Escherichia coli* by co-expressing AjCGT, AjOMT and upstream genes, and achieved *de novo* biosynthesis of 4-OMGA-Glc through plasmid expression. Bergenin could be obtained after acid treatment. Finally, the production of bergenin could reach 1.41 g/L in a 3-L bioreactor. This work provides a complete story and demonstrates the power of synthetic biology in preparing complex natural products. However, studies on biochemistry, structure and function of key enzyme are relatively weak. The catalytic mechanisms of 2CGT and OMT should be investigated to improve novelty and quality of this work. Otherwise, the reviewer would not recommend this work for publication in *Nat. Commun.*

We are greatly appreciated for your constructive comments and helpful suggestions and pointing out the issues to which we should pay attention. Following these suggestions, the catalytic mechanisms of 2CGT and OMT were further investigated, respectively, to improve novelty and quality of our work. We have also carefully checked and revised our manuscript. The responses are listed as follows.

Major concerns:

1. What's the major difference between AjCGT and previously reported CGTs? While AjCGT shares highly similarity with UGT708 family glycosyltransferases (as shown in Fig. S1), the authors claim that 'Different C-glycosylation property from other reported plant CGTs, the two CGTs displayed the ability to introduce C-sugar into the aromatic ring with acyl 3,4,5-pyrogallol structure, suggesting that they might be a novel type of plant CGT'. However, the evidence provided is limited. Whether other CGTs could catalyze 2-C-glycosylation of **2** or **3** needs

to be investigated.

Response #2-1:

Thanks a lot for your very constructive suggestions. Following the suggestion, seven representative CGTs including FeCGTb¹, UGT708D1², MiCGT³, TcCGT1⁴, PIUGT43⁵, OsCGT⁶ and AbCGT⁷ were selected to investigate the C-glycosylation activity to GA (2) and 4-OMGA (3). According to the results of HPLC-MS/MS², all the seven CGTs showed no C-glycosylation activity to GA (2) and 4-OMGA (3) with the acyl 3,4,5-pyrogallol structure, respectively (Figures r2-1 and r2-2).

Figure r2-1. Probing the C-glycosylation activities of the representative reported CGTs (FeCGTb¹, UGT708D1², MiCGT³, TcCGT1⁴, PIUGT43⁵, OsCGT⁶ and AbCGT⁷) to GA (2). All these seven CGTs showed no C-

glycosylation activity to GA (2). (Supplementary Fig. 1 in the new version)

Figure r2-2. Probing the C-glycosylation activities of the representative reported CGTs (FeCGTb, UGT708D1, MiCGT, TcCGT1, PIUGT43, OsCGT and AbCGT) to 4-OMGA (3). All these seven CGTs showed no C-glycosylation activity to 4-OMGA (3). (Supplementary Fig. 2 in the new version)

In addition, we also investigated the C-glycosylation activity of AjCGT1 to the native substrates (such as phloretin, 2-hydroxynaringenin, maclurin, apigenin, and 5,7-dihydroxycoumarin as shown in **Scheme 1**) of reported CGTs, respectively, which is also suggested by the first reviewer. AjCGT1 also showed C-glycosylation activity to these substrates (**Scheme 1**) with acyl phloroglucinol structure. The HPLC analysis results were shown in **Figures r2-3–r2-7**. We have added these detailed results in the manuscript and **Supplementary**

Material in the new version (**Supplementary Fig. 10–14**). Therefore, AjCGT1 is a C-glycosyltransferase that not only recognized substrates with acyl phloroglucinol structure but also could catalyze unusual substrates having 3,4,5-trihydroxybenzoic acid-like structure.

Scheme2-1. Probing the catalytic activity of AjCGT1 with substrates having 2, 4, 6-trihydroxyacetophenone-like structure.

Figure r2-3. HPLC-MS/MS² analysis of AjCGT1 catalyzing the C-glycosylation of phloretin. a) AjCGT1 catalyzed the C-glycosylation of

phloretin; b) HPLC chromatograms of the AjCGT1 catalyzing reaction, control and the standard C-glycosides; c) Typical negative ion MS and MS² spectra for the C-glycosylated products **p1** and **p2**.

Figure r2-4. HPLC-MS/MS² analysis of AjCGT1 catalyzing the C-glycosylation of 2-hydroxynaringenin. a) AjCGT1 catalyzed the C-glycosylation of 2-hydroxynaringenin; b) HPLC chromatograms of the AjCGT1 catalyzing reaction, control and the standard C-glycosides; c) Typical positive ion MS and MS² spectra for the C-glycosylated products **p3/p4**.

Figure r2-5. HPLC-MS/MS² analysis of AjCGT1 catalyzing the C-glycosylation of maclurin. a) AjCGT1 catalyzed the C-glycosylation of maclurin; b) HPLC chromatograms of the AjCGT1 catalyzing reaction, control and the standard C-glycosides; c) Typical negative ion MS and MS² spectra for the C-glycosylated product **p5**.

Figure r2-6. HPLC-MS/MS² analysis of AjCGT1 catalyzing the C-glycosylation of apigenin. a) AjCGT1 catalyzed the C-glycosylation of apigenin; b) HPLC chromatograms of the AjCGT1 catalyzing reaction and the control; c) Typical negative ion MS and MS² spectra for the C-glycosylated product **p6** and O-glycosylated product **p7**.

Figure r2-7. HPLC-MS/MS² analysis of AjCGT1 catalyzing the C-glycosylation of 5,7-dihydroxycoumarin. a) AjCGT1 catalyzed the C-glycosylation of 5,7-dihydroxycoumarin; b) HPLC chromatograms of the AjCGT1 catalyzing reaction, control and the standard C-glycoside; c) Typical negative ion MS and MS² spectra for the C-glycosylated product **p8**.

2. It would be valuable to investigate the relationship between the sequence and structure of AjCGT. Crystal structure analysis and theoretical calculation would provide important approaches to dissect the mechanisms.

Response #2-2:

Many thanks for your great suggestions. After many efforts, we failed to obtain the crystal structure of AjCGT. Accordingly, to further explore the catalytic

mechanisms AjCGT1 in recognizing substrates with 3,4,5-trihydroxybenzoic acid-like structure, GA (**2**) and 4-OMGA (**3**) were docked into AjCGT1 with the guidance of crystal structures of UGT708C1 (6LLZ) and GgCGT (6L7H) (**Figure r2-8**). 14 candidate active sites (H23, H27, L84, N85, W92, V121, T142, F147, W151, F188, H367, G368, D369 and Q370) were found around the acceptors (**2** and **3**). Alanine-scanning of the 14 candidate active sites showed 10 mutants (H23A, H27A, W92A, T142A, F147A, W151A, F188A, H367A, G368A, D369A and Q370A) greatly lost their activities (**Figure r2-9**).

Figure r2-8. Exploring the catalytic sites of AjCGT1 by protein modelling and molecular docking. The protein was modelled using Swiss-Model. GA (**2**) and 4-OMGA (**3**) were docked into the active sites using AutoDock Vina. GA (**2**), 4-OMGA (**3**) and UDP-Glc are shown in yellow and green, respectively. a) Substrate-binding pocket of AjCGT1 for GA (**2**) and amino acid residues surrounding GA (**2**). b) Substrate binding pocket of AjCGT1 for 4-OMGA (**3**) and amino acid residues surrounding 4-OMGA (**3**). (**Supplementary Fig 15** in the new version)

Figure r2-9. Alanine-scanning of the candidate active sites of AjCGT1. The C-glycosylation activities of the mutants were investigated by GA (2), 4-OMGA (3) and phloretin (Supplementary Fig 16 in the new version).

Figure r2-10. Sequences alignment of AjCGT1 and other reported CGTs. Besides the highly conserved His23, another two basic amino acid residues His27 and W92 which are different from the reported CGTs were found among the 14 candidate active sites.

Further sequences and structures alignment of AjCGT1 and other CGTs showed that another two basic amino acid residues His27 and W92 were found near the highly conserved His23 (Figures r2-10 and r2-11). The conserved His23 is generally considered as a necessary active site to specifically deprotonate the hydroxyl group of the substrate and makes it a nucleophile to attack the sugar donor UDP-Glc^{4,8,9} (Figure r2-11). The mutation of conserved His23 usually disrupts the C-glycosylation activity of CGTs. Just like the other reported CGTs, the mutant of AjCGT1-H23A completely lost its activity. The mutant of AjCGT1-H27A only partially loss of the activities in glycosylating GA

(2) and 4-OMGA (3) as well as phloretin (**Figure r2-9**). Especially for AjCGT1-W92A, its C-glycosylation activity to GA (2) and 4-OMGA (3) were completely lost, while that to phloretin still remained about 15% (**Figure r2-9**). Therefore, the basic sites His27 and W92 might also act as general bases to deprotonate the hydroxyl group of gallic acids to assist His23 complete the C-glycosylation of gallic acids (**Figure r2-11**). In addition, some active sites in the binding pocket also make GA (2) and 4-OMGA (3) in the proper orientation. This information about the catalytic mechanism of AjCGT1 was added in the main text.

Figure r2-11. The proposed catalytic mechanism of AjCGT1 and other reported CGTs^{4,8,9}. a) The catalytic mechanism of known CGTs in C-glycosylation of substrates having 2, 4, 6-trihydroxyacetophenone-like structure; b) The proposed catalytic mechanism of AjCGT1 in C-glycosylation of GA (2); c) The key basic amino acid residues of AjCGT1 (green), UGT708C1 (blue) and GgCGT (brown).

Minor concerns:

1. The low affinity ($K_m=279.6 \mu\text{M}$) of AjCGT1 towards **3** is unexpected. It seems to indicate that **3** is not the endogenous substrate for AjCGT1. Have the authors used **2** as sugar acceptor to measure the kinetic parameter of AjCGT1?

Response #2-M-1:

Thank you very much for your expertise. Yes, the apparent K_m of AjCGT1 towards **3** revealing the low affinity, which is also mentioned by the first reviewer. We attempted to measure the K_m of AjCGT1 towards **2** many times and still cannot obtain the data to calculate the kinetic parameters. After checking the literatures, it was reported that gallic acid compounds are not only very unstable under neutral and alkaline conditions, but also easily undergoes non-specific binding with proteins due to its pyrogallol structure.^{10,11} According to the results of our stability experiments, GA (**2**) was gradually decreased in neutral and alkaline buffers such as Tris-HCl and phosphate buffers, and it even completely disappeared within 10 minutes in phosphate buffer (**Figure r2-12** and **Figure r2-13**). In addition, the 4-OMGA (**3**) was also slightly decreased in buffer solutions (**Figure r2-14**). Thus, the apparent kinetic parameters of AjCGT1 we obtained here may not be its real kinetic parameters and can't reflect its enzymatic property including the affinity. We have added this information to the main text to make the description clear. These figures were also added to the **Supplementary Material** (new version) as **Supplementary Fig. 5–7**.

Figure r2-12. Investigation of the stability of GA (2) in different buffers. GA (2) was added into MeOH (control) and the corresponding buffers for 10 minutes to evaluate its stability. HPLC analysis results showed that the peak areas of GA (2) in MeOH, pH7.0 Tris-HCl, pH8.0 Tris-HCl, pH7.0 Na₂HPO₄-NaH₂PO₄ and pH8.0 Na₂HPO₄-NaH₂PO₄ buffer were 548 (control), 513, 412, 495 and 0, respectively. These results indicated GA (2) could be spontaneously decreased even in a short reaction time (10 min).

Figure r2-13. Investigation of the stability of GA (2) at different incubation times. GA (2) was added into pH7.0 $\text{Na}_2\text{HPO}_4\text{-NaH}_2\text{PO}_4$ buffer and was incubated for 10, 20, 30, and 45 min, respectively. HPLC analysis results showed that the peak areas of GA (2) are 548 (control), 495 (10 min), 429 (20 min), 408 (30min) and 315 (45 min), respectively. These results indicated that GA (2) was unstable in phosphate buffer solution.

Figure r2-14. Investigation of the stability of 4-OMGA (3) at different incubation times. 4-OMGA (3) was added into pH 8.0 Na₂HPO₄-NaH₂PO₄ buffer and was incubated for 0, 10, 20, 30, and 60 min, respectively. HPLC analysis results showed that the peak areas of 4-OMGA (3) are 1284 (control), 1174 (0 min), 1149 (10 min), 1129 (20 min), 1095 (30min) and 954 (60 min), respectively. These results indicated that 4-OMGA (3) was unstable in phosphate buffer solution.

2. Line 173, the authors mentioned that the relative rate of AjOMT2 catalyzing **6** to generate **1** is low. The reviewer did not find obvious difference between Fig 2k and Fig 2n.

Response #2-M-2:

Thanks for your kind reminding, and we were very sorry for such mistake. We have carefully rechecked the conversion rates of AjOMT2 catalyzing GA (**2**) and norbergenin (**6**). The relative rate of AjOMT2 catalyzing **2** to generate **3** was 62.4%±3.5%, the relative rate of AjOMT2 catalyzing **6** to generate **1** was

72.2%±5.3% in **Fig. 2**. As you said, there is no significant difference between Fig 2k and Fig 2n in conversion rates. We have revised the description and added the detailed conversion rates in the manuscript.

3. The catalytic efficiencies of AjCGT2 and AjOMT3 towards different acceptors should be measured, respectively.

Response #2-M-3:

Thank you so much for your expertise suggestion. The catalytic efficiency of AjCGT2 toward 4-OMGA (**3**) as well as AjOMT3 toward GA (**2**) and norbergenin (**6**) were also measured, respectively. As the enzymatic parameter can't reveal the catalytic efficiency of AjCGTs and AjOMTs due to the instability of substrates, the conversion rates of AjCGTs and AjOMTs in the same reaction time were used to compare the efficiencies. The results are shown in **Table r2-1**, **Figures r2-15–r2-18**. The corresponding data was also supplemented in the **Supplementary Fig. 8, 9, 24 and 25** in the new version of the manuscript.

Table r2-1 The catalytic activity of AjCGT1, AjCGT2, AjOMT2 and AjOMT3 to different substrates.

Enzymes	Conversion rates of substrates		
	GA (2)	4-OMGA (3)	Norbergenin (6)
AjCGT1	45.8%	61.5%	/
AjCGT2	29.8%	47.0%	/
AjOMT2	62.4%	/	72.2%
AjOMT3	10.6%	/	46.4%

Figure r2-15. The catalytic activity of AjCGT1 and AjCGT2 with GA (2) as an acceptor. a) AjCGT1 and AjCGT2 catalyzing the C-glycosylation of GA (2). b) HPLC chromatograms of the AjCGT1 and AjCGT2 catalyzing reactions and control. c) Typical negative ion MS and MS² spectra for the C-glycosylated product 5.

Figure r2-16. The catalytic activity of AjCGT1 and AjCGT2 with 4-OMGA (3) as an acceptor. a) AjCGT1 and AjCGT2 catalyzing the C-glycosylation of 4-OMGA (3). b) HPLC chromatograms of the AjCGT1 and AjCGT2 catalyzing reactions and control. c) Typical negative ion MS and MS² spectra for the C-glycosylated product 4.

Figure r2-17. The catalytic activity of AjOMT2 and AjOMT3 with GA (2) as an acceptor. a) AjOMT2 and AjOMT3 catalyzing the O-methylation of GA (2). b) HPLC chromatograms of the AjOMT2 and AjOMT3 catalyzing reactions and control. c) UV absorption spectrum and typical positive ion MS spectra for the product 3.

Figure r2-18. The catalytic activity of AjOMT2 and AjOMT3 with norbergenin (**6**) as an acceptor. a) AjOMT2 and AjOMT3 catalyzing the O-methylation of norbergenin (**6**). b) HPLC chromatograms of the AjOMT2 and AjOMT3 catalyzing reactions and control. c) UV absorption spectrum and typical positive ion MS spectra for the product **1**.

4. There seem to be some mistakes in the ligands labeled in Figures S4, S5, S8, and S9. The authors should label which buffer was used for the enzyme property studies in the figures.

Response #2-M-4:

Thanks for your kind reminding. We have revised the mistakes in the legends of Figures S4, S5, S8 and S9 (**Figures S17, 18, 28 and 29** in the new

version). In addition, the buffers used for the enzyme property studies were also indicated in the legends.

5. Figure S4, AjCGT1 shows a relative conversion rate of approximately 15% and 80% in Buffer A (pH=7) and Buffer B (pH=7), respectively. Please confirm it.

Response #2-M-5:

Thank you so much for your expertise. Yes, the catalytic activity of AjCGT1 exhibited large difference in Buffer A ($\text{Na}_2\text{HPO}_4\text{-NaH}_2\text{PO}_4$ buffer, pH 7.0) and Buffer B (Tris-HCl buffer, pH 7.0) towards GA (2), and thanks a lot for your kind reminding. The main reason for the large difference is the instability of GA (2) in the buffer of $\text{Na}_2\text{HPO}_4\text{-NaH}_2\text{PO}_4$ pH 7.0 as shown in **Figures r2-13**. To reduce the effect of the instability of GA (2), we shortened the post-processing time and HPLC detecting time and obtained a new **Figure S4 (Figure r2-19, Supplementary Figure 17 in the new version)**

Figure r2-19. Effects of pH buffers, temperatures, and divalent metal ions on the activity of AjCGT1. a) Effects of various pH buffers (pH 5.0–6.0, citric acid-sodium citrate buffer; pH 6.0–7.0, $\text{Na}_2\text{HPO}_4\text{-NaH}_2\text{PO}_4$ buffer; pH 7.0–10.0, Tris-HCl buffer; pH 10.0–11.0, $\text{Na}_2\text{CO}_3\text{-NaHCO}_3$ buffer); b) Effects of various temperatures; c) Effects of various divalent metal ions. GA (2) was used as the acceptor.

References

1. Nagatomo, Y. et al. Purification, molecular cloning and functional characterization of flavonoid C-glucosyltransferases from *Fagopyrum esculentum* M. (buckwheat) cotyledon. *Plant J.*, **80**, 437–448 (2014).
2. Hirade, Y. et al. Identification and functional analysis of 2-hydroxyflavanone C-glucosyltransferase in soybean (*Glycine max*). *FEBS Lett.*, **589**, 1778–1786 (2015).
3. Chen, D. et al. Probing the catalytic promiscuity of a regio- and stereospecific C-glycosyltransferase from *Mangifera indica*. *Angew. Chem. Int. Ed.* **127**, 12869–12873 (2015).
4. He, J. et al. Molecular characterization and structural basis of a promiscuous C-glycosyltransferase from *Trollius chinensis*. *Angew. Chem. Int. Ed.*, **58**, 11513–11520 (2019).
5. Wang, X., Li, C., Zhou, C., Li, J. & Zhang, Y. Molecular characterization of the C-glycosylation for puerarin biosynthesis in *Pueraria lobata*. *Plant J.*, **90**, 535–546 (2017).
6. Brazier-Hicks, M. et al. The C-glycosylation of flavonoids in cereals. *J. Biol. Chem.*, **284**, 17926–17934 (2009).
7. Xie, K., Zhang, X., Sui, S., Ye, F. & Dai, J. Exploring and applying the substrate promiscuity of a C-glycosyltransferase in the chemo-enzymatic synthesis of bioactive C-glycosides. *Nat. Commun.* **11**, 5162 (2020).
8. Zhang, M. et al. Functional characterization and structural basis of an efficient di-C-glycosyltransferase from *Glycyrrhiza glabra*. *J. Am. Chem. Soc.*, **142**, 3506–3512 (2020).
9. Gutmann, A. & Nidetzky, B. Switching between O- and C-glycosyltransferase through exchange of active-site motifs. *Angew. Chem. Int. Ed.* **51**, 12879–12883 (2012).
10. Zhang, J., et al. Changes in composition and enamel demineralization inhibition activities of gallic acid at different pH values. *Acta Odontol. Scand.*

73, 595–601 (2015).

11. Nascimento, J. M. D., et al. Evaluation of the influence of temperature on the protein-tannic acid complex. *Int. J. Biol. Macromol.* **182**, 2056–2065 (2021).

Response to Reviewer #3

Comments:

The manuscript by Yan et al. elucidated the biosynthetic pathway of bergenin in *Ardisia japonica* and constructed the *de novo* biosynthetic pathway of the direct precursor of bergenin. Although this work established a foundation for biosynthesis of bergenin, the novelty of this research is not enough and the writing also is careless.

We would like to greatly appreciate you for your comments and suggestions. Bergenin is the main bioactive ingredient of the traditional Chinese medical plant *Ardisia japonica* and has been used in the clinical treatment of chronic bronchitis and pulmonary tuberculosis. However, its low abundance in nature and structural complexity hampers the accessibility through traditional crop-based manufacturing or chemical synthesis. Synthetic biology provides an opportunity to solve the drug source problem of bergenin. However, the biosynthetic pathway of bergenin was unknown especially for the process of C-sugar and O-methyl group introduction. What's more, due to the unique structure of bergenin, it's difficult to utilize known genetic elements to achieve the *de novo* its biosynthesis.

In the present work, we firstly unraveled the biosynthetic pathway of bergenin by decrypting the key novel highly regio- and/or stereo-selective 2-C-glycosyltransferases and 4-O-methyltransferases. Then, for the first time, we designed and rebuilt in *Escherichia coli* the *de novo* biosynthetic pathway of the direct precursor of bergenin. The production of bergenin was greatly improved to reach 1.41 g L⁻¹ through further metabolic engineering. This work has promising potential for large-scale production of bergenin instead of plant extract, and will supply bergenin *via* a microbial factory.

We are very sorry for the careless writing in the previous version and thanks for your kind reminding. We have carefully checked and revised our manuscript in the new version. Please find the revising items in the corresponding responses.

1. The distribution of bergenin in medicinal plants is very wide, but no reference was provided. It is necessary to discuss the possibility of the synthetic pathway of bergenin from other species.

Response #3-1:

Thanks a lot for your helpful suggestions. According to the suggestion, some relevant references (ref 1-3; ref 7-9 in the new version of the main text) about the distribution of bergenin in medicinal plants were provided in the manuscript. The possibility of the synthetic pathway of bergenin from *Saxifraga stolonifera* was introduced in the section of “**Introduction**” (ref 4; ref 17 in the new version of the main text).

References:

1. Deng, J., Xiao, X., Tong, X. & Li, G. Preparation of bergenin from *Ardisia crenata* Sims and *Rodgersia sambucifolia* Hemsl based on microwave-assisted extraction/high-speed counter-current chromatography. *Sep. Purif. Technol.* **74**, 155–159 (2010).
2. Li, P., Yang, S. & Zeng, Y. Advances in studies on resources for medicinal source plants with bergenin. *Chin. Tradit. Herb. Drugs* **40**, 1500–1505 (2009).
3. Sun, X., et al. Comparative studies on content of arbutin, bergenin and catechin in different part of *Bergenia purpurascens* and *B. crassifolia*. *China J Chin. Mater. Med.* **35**, 2079–2082 (2010).
4. Taneyama, M., Yoshida, S. Studies on C-glycosides in higher plants II. Incorporation of ¹⁴C-glucose into bergenin and arbutin in *Saxifraga stolonifera*. *Bot. Mag. Tokyo* **92**, 69–73 (1979).

Line59 (new version) “...¹⁴C-glucose incorporation experiment in *Saxifraga stolonifera* leaves also indicated GA might be a glucosyl acceptor in bergenin biosynthesis¹⁷...”

2. It is important to screen the function of other OMT genes which co-expressed

with the bait genes GTs.

Response #3-2:

Many thanks for your expertise suggestions. Besides *AjOMT2* and *AjOMT3*, the function of *AjOMT4–7* genes which also co-expressed with the bait genes of *AjCGT1* and *AjCGT2* was investigated. The results of *in vitro* enzyme catalytic activity indicated that *AjOMT4–7* showed relatively low or no catalytic activity towards GA (2) and norbergenin (6). The catalytic activities of these *AjOMTs* were shown in **Figures r3-1** and **Figure r3-2**.

Figure r3-1. The catalytic O-methylation activity of AjOMTs towards GA (2). a) O-methylation of GA (2) reactions catalyzed by AjOMTs; b) HPLC analysis of the enzymatic reactions. (**Supplementary Fig. 26** in the new version)

Figure r3-2. The catalytic O-methylation activity of AjOMTs towards norbergenin (6). a) O-methylation of norbergenin (6) reactions catalyzed by AjOMTs; b) HPLC analysis of the enzymatic reactions. (**Supplementary Fig. 27** in the new version)

3. “the relatively higher conversion rates of AjCGT1 than those of AjCGT2 with GA (2) and 4-OMGA (3)”. But in Fig2, there was no significant difference.

Response #3-3:

Thank you very much for your carefulness. Yes, due to the small size of the chromatograms in Fig2, the difference in conversion rates between AjCGT1

and AjCGT2 were not observed obviously. In fact, the conversion rates of AjCGT1 for GA (**2**) and 4-OMGA (**3**) were 45.8%±5.2%, 61.5%±6.5% respectively, and those of AjCGT2 for GA (**2**) and 4-OMGA (**3**) were 29.8%±4.5%, 47.0%±5.7% respectively (**Figure r3-3** and **r3-4**). To make the information clear, the conversion rates of AjCGT1 and AjCGT2 with GA (**2**) and 4-OMGA (**3**) were also provided in the manuscript and provided as **Supplementary Fig. 8** and **9**.

Figure r3-3. The catalytic activity of AjCGT1 and AjCGT2 with GA (2**) as an acceptor.** a) AjCGT1 and AjCGT2 catalyzing the C-glycosylation of GA (**2**). b) HPLC chromatograms of the AjCGT1 and AjCGT2 catalyzing reactions and control. c) Typical negative ion MS and MS² spectra for the C-glycosylated product **5**.

Figure r3-4 The catalytic activity of AjCGT1 and AjCGT2 with 4-OMGA (3) as an acceptor. a) AjCGT1 and AjCGT2 catalyzing the C-glycosylation of 4-OMGA (3). b) HPLC chromatograms of the AjCGT1 and AjCGT2 catalyzing reactions and control. c) Typical negative ion MS and MS² spectra for the C-glycosylated product 4.

4. In Supplementary Fig. 4, 5, 8 and 9, (a) and (b) of these graphs do not correspond to the headings.

Response #3-4:

Thanks so much for your kind reminding. We have revised the figure legends (**Supplementary Fig. 17, 18, 28 and 29** in the new version) and carefully checked all through the manuscript.

5. When added the codon-optimized strategy, the strain Y4 harboring codon-optimized AjOMT2*^{opt} and AjCGT1^{opt} (codon-optimized strain M1) showed the highest catalytic activity, not codon-optimized strain M2 (carrying pET-AjOMT2-AjCGT1*).

Response #3-5:

Thanks a lot for your expertise. We repeated the experiment for more than three times again, and obtained the same results. Codon optimization of recombinant enzymes usually could increase their expression levels in host cells and the catalytic activities. The degree of improvement in enzyme activity may be different depending on the type of enzymes involved, which means the codon optimization was more effective in improvement of the catalytic activity of AjOMTs than that of AjCGTs. Thus, the activity of codon-optimized AjOMT2*^{opt} and AjCGT1^{opt} may increase significantly.

6. There is wrong description in Fig. 4h.

Response #3-6:

Thank you so much for your carefulness. We have corrected the description in Fig. 4h. *AjOMT2** was revised to *AjOMT2^{opt}* and *AjCGT1** was changed to *AjCGT1^{opt}*.

7. The obtained yields of bergenin from *E. coli* exhibit promising results; however, it is essential to compare it to the yields achieved through analogous engineering methodologies for related compounds.

Response #3-7:

Thanks a lot for such helpful suggestions. Following your suggestion, the information that comparing the yields with other related compounds is important and should be provided. We have added the yields of the representative C-glycosides (puerarin, isovitexin, vitexin and apigenin di-C-arabinoside)¹⁻⁴ produced by *de novo* biosynthesis. Most of the yields can reach milligram level. Up to date, the highest reported yield was obtained in the biosynthesis of isovitexin, which was 206 mg L⁻¹. As suggested, this information was supplemented in the section of “Fed-batch production of 4-OMGA-Glc in bioreactor”.

References:

1. Liu, Q. et al. *De novo* biosynthesis of bioactive isoflavonoids by engineered yeast cell factories. *Nat. Commun.* **12**, 6085 (2021).
2. Vanegas, K. G., et al. Indirect and direct routes to C-glycosylated flavones in *Saccharomyces cerevisiae*. *Microb. Cell Fact.* **17**, 107 (2018).
3. Chen, Z., et al. *De novo* biosynthesis of C-arabinosylated flavones by utilization of indica rice C-glycosyltransferases. *Bioresour. Bioprocess.* **8**, 49 (2021).
4. Chong, Y, et al. Production of four flavonoid C-glucosides in *Escherichia coli*. *J. Agric. Food Chem.* **71**, 5302–5313 (2023).

Finally, we would like to again thank all the reviewers for the excellent review and supportive suggestions! We trust you'll agree that our work & manuscript have been greatly improved with your guidance. Many thanks!

Reviewers' Comments:

Reviewer #1:

Remarks to the Author:

I think this study is an important achievement leading to the microbial production of bergenin, as I mentioned in my previous review. In this revised version, the authors have mostly appropriately revised the manuscript in accordance with the reviewers' comments, and the manuscript has been significantly improved.

The authors performed the enzyme reaction of several CGTs and found that the glycosylation of GA and 4-OMGA is characteristic reaction for AjCGT1 and AjCGT2. This result suggests that AjCGT1 and AjCGT2 may be involved in the biosynthesis of this compound, although their activity is quite low. They also performed the enzyme reaction of AjCGT1 with other substrates to reveal the substrate preference of AjCGT1, which provides useful information for understanding the nature of the enzyme.

However, there remains some concerns.

I suggest again some improvement of this manuscript.

1) I think evaluation of the enzyme activity is still insufficient. The data presented in for the substrate specificity are qualitative but not quantitative, and the reaction condition (20 µg, 1 hr) (L. 571–573) seems to be saturated (For ex., Supplementary Fig. 9, 10, etc.).

Please present the enzymatic parameters (at least the specific activity) with these substrates under non-saturated condition (initial rate) and compare them.

The authors explained that the reaction parameters cannot be estimated correctly owing to the instability of the substrate at pH 8.0 in phosphate buffer, but the reaction condition was described as pH 7.0 in Tris buffer (L. 571–573), which is more stable. In addition, 4-OMGA is much more stable than GA (Supplementary Fig. 7).

If the substrate is unstable, it should be performed in a shorter time such as 1 min with an increased amount of enzyme. Then the activity can be evaluated in comparison with other substrates or the reported activities of other related enzymes.

To begin with, if the reaction is not completed in a few minutes with this amount of enzyme, I think the activity is weak compared to other CGTs.

In addition, the conditions for each reaction are not clear. If the reaction conditions are different, please explain it such as in the figure legend in the supplementary figures.

2) The enzymatic activity of AjOMT2 has not been clearly demonstrated.

It is necessary to check the enzymatic parameters (specific activity) as well and compare it with other enzymes.

Please discuss on the role of the enzymes in nature, with the enzymatic parameters.

Reviewer #2:

Remarks to the Author:

The authors have carefully revised the manuscript. My queries have been fully addressed. The revised manuscript has remarkably improved quality. Thus, I suggest it be published in its current form.

Reviewer #3:

Remarks to the Author:

All of my concerns have been answered.

To Reviewer #1:

Reviewer comments:

I think this study is an important achievement leading to the microbial production of bergenin, as I mentioned in my previous review. In this revised version, the authors have mostly appropriately revised the manuscript in accordance with the reviewers' comments, and the manuscript has been significantly improved.

We would like to thank you very much to review our manuscript again and give us such invaluable suggestions and comments, which are very helpful to improve the quality of our work and manuscript. Following your suggestions, we have addressed each point below.

The authors performed the enzyme reaction of several CGTs and found that the glycosylation of GA and 4-OMGA is characteristic reaction for AjCGT1 and AjCGT2. This result suggests that AjCGT1 and AjCGT2 may be involved in the biosynthesis of this compound, although their activity is quite low. They also performed the enzyme reaction of AjCGT1 with other substrates to reveal the substrate preference of AjCGT1, which provides useful information for understanding the nature of the enzyme.

However, there remains some concerns. I suggest again some improvement of this manuscript.

1) I think evaluation of the enzyme activity is still insufficient. The data presented in for the substrate specificity are qualitative but not quantitative, and the reaction condition (20 µg, 1 hr) (L. 571–573) seems to be saturated (For ex., Supplementary Fig. 9, 10, etc.).

Please present the enzymatic parameters (at least the specific activity) with these substrates under non-saturated condition (initial rate) and compare them.

The authors explained that the reaction parameters cannot be estimated correctly owing to the instability of the substrate at pH 8.0 in phosphate buffer, but the reaction condition was described as pH 7.0 in Tris buffer (L. 571–573),

which is more stable. In addition, 4-OMGA is much more stable than GA (Supplementary Fig. 7).

If the substrate is unstable, it should be performed in a shorter time such as 1 min with an increased amount of enzyme. Then the activity can be evaluated in comparison with other substrates or the reported activities of other related enzymes.

To begin with, if the reaction is not completed in a few minutes with this amount of enzyme, I think the activity is weak compared to other CGTs.

In addition, the conditions for each reaction are not clear. If the reaction conditions are different, please explain it such as in the figure legend in the supplementary figures.

Response #1-1

We are greatly appreciated for the constructive comments and helpful suggestions. Following these suggestions, the enzyme activities of AjCGT1 and AjCGT2 were further evaluated and we have also added these results to the manuscript.

As suggested, we have re-calculated the specific activity of AjCGT1 and AjCGT2 with substrates of 4-OMGA and GA under non-saturated condition in pH 7.0 Tris-HCl buffer as shown in **Table R1** (Supplementary Table 9 in the new version of the article).

AjCGT1 exhibited higher specific activity to 4-OMGA and GA in comparison with that of AjCGT2. In addition, to compare the enzymatic activity of AjCGT1 with that of the representative reported CGTs, the stable substrate phloretin was used as the common substrate. According to the experimental results, AjCGT1 indeed showed lower specific activity comparing with that of FeCGT while higher specific activity with that of OsCGT and CuCGT.

Table R1 The specific activity of AjCGT1, AjCGT2 and other reported CGTs

Enzymes	Substrates	specific activity (pkat/mg) ^a
AjCGT1	4-OMGA (3)	3667
AjCGT2	4-OMGA (3)	1871
AjCGT1	GA (2)	868
AjCGT2	GA (2)	370
AjCGT1	phloretin	13827
OsCGT ^[1]	phloretin	587
FeCGT ^[2]	phloretin	34200
CuCGT ^[2]	phloretin	6600

a: The reaction was performed with 0.4 mM acceptors, 0.8 mM UDP-Glc and 10 µg purified AjCGTs in a total volume of 100 µL and the mixture was incubated at 40 °C for 10 min. For the acceptor of 4-OMGA, the reaction was performed in 50 mM Na₂HPO₄-NaH₂PO₄ buffer (pH 8.0) while for GA, the reaction was performed in 50 mM Tris-HCl buffer (pH 7.0).

Yes, 4-OMGA is much more stable than GA. According to the suggestion, the kinetic parameters of AjCGT1 was measured in a shorter reaction time (1 min) with an increased amount of enzyme (50 µg) (**Fig R1** and **Table R2**). Under these conditions, the K_{cat} was higher than that measured under previous reaction conditions. The K_{cat}/K_m of AjCGT1 was indeed lower than that of other reported CGTs with the stable substrates. Therefore, to further compare the enzymatic activity, phloretin was also used to measure the kinetic parameters of AjCGT1 (**Fig R2**). AjCGT1 exhibited similar K_{cat}/K_m with AbCGT but lower enzymatic activity comparing the reported OsCGT and FeCGT (**Table R3**).

Therefore, AjCGT1 indeed showed a lower enzymatic activity to the reported CGTs with the same substrate.

Fig R1 Determination of kinetic parameters for recombinant AjCGT1 with substrate of 4-OMGA (**3**). UDP-Glc was used as the sugar donor. The reactions were performed with a short reaction time (1 min) and an increased amount of enzyme (50 μg) in a total volume of 100 μL to avoid the effect of the instability of 4-OMGA (**3**).

Table R2 The kinetic parameters of AjCGT1 with the substrate of 4-OMGA

Enzyme	Substrate	V_{\max} ($\text{nmol}\cdot\text{min}^{-1}\cdot\text{mg}^{-1}$)	K_m (μM)	K_{cat} (s^{-1})	K_{cat}/K_m ($\text{M}^{-1}\cdot\text{s}^{-1}$)
AjCGT1	4-OMGA	368.8	265.3	0.31	1154.55

Fig R2 The kinetic parameters of AjCGT1 with the substrate of phloretin. UDP-Glc was the sugar donor. The reaction was performed with the reaction time of 10 min and the enzyme amount of 1 μg in a reaction system of 100 μL .

Table R3 The kinetic parameters of AjCGT1 with the substrate of 4-OMGA

Enzymes	Substrates	K_m (μM)	K_{cat} (s^{-1})	K_{cat}/K_m ($\text{M}^{-1}\cdot\text{s}^{-1}$)
AjCGT1	phloretin	40.59	0.69	1.7×10^4
OsCGT ^[1]	phloretin	4.78	10.84	2.3×10^6
FeCGT ^[2]	phloretin	$<0.5^c$	12	$>2.4\times 10^7$
AbCGT ^[3]	phloretin	146	2	1.4×10^4

In addition, to make the statement clear, the reaction conditions for each reaction have been added in the figure legends in the Supplementary Figures 8–14 as suggested.

2) The enzymatic activity of AjOMT2 has not been clearly demonstrated.

It is necessary to check the enzymatic parameters (specific activity) as well and compare it with other enzymes.

Please discuss on the role of the enzymes in nature, with the enzymatic parameters.

Response #1-2:

Thanks for your professional advice. We have checked the enzymatic parameters of AjOMT2 and compared that with other similar methyltransferases such as CrOMT1, VpOMT4 and VpOMT5, which showed methylation activity to myricetin and tricetin, respectively. Comparing with other similar OMTs, AjOMT2 showed a moderate value ($250 \text{ M}^{-1} \text{ s}^{-1}$) of K_{cat}/K_m (Table R4, Supplementary Table 9 in the new version of the article). We have also added the discussion about the role of these enzymes in nature with the enzymatic parameters in manuscript as suggested.

Table R4 The kinetic parameters of OMTs with different substrates

Enzymes	Substrates	K_m (μM) ^a	K_{cat} (s^{-1}) ^a	K_{cat}/K_m ($\text{M}^{-1}\cdot\text{s}^{-1}$) ^a
AjOMT2 ^a	GA (2)	72.0	1.8×10^{-2}	250
AjOMT2 ^a	norbergenin (6)	493.6	1.9×10^{-2}	38.5
CrOMT1 ^[4]	myricetin	23.8	6.7×10^{-4}	28.0
CrOMT1 ^[4]	tricetin	0.6	7.6×10^{-4}	1387
VpOMT4 ^[5]	tricetin	158	1.11×10^{-2}	70
VpOMT5 ^[5]	tricetin	109	9.13×10^{-2}	838

a: The kinetic parameters of AjOMT2 were measured at 40 °C and pH 7.0 (50 mM Tris-HCl buffer).

Thanks again for the helpful and professional suggestions.

References:

- [1] Ferreyra, M. L. F. et al. Identification of a bifunctional maize C- and O-glycosyltransferase. *J. Biol. Chem.*, **288**, 31678–31688 (2013)
- [2] Ito, T., Fujimoto, S., Suito, F., Shimosaka, M., & Taguchi, G. C-glycosyltransferases catalyzing the formation of di-C-glycosyl flavonoids in citrus plants. *Plant J.*, **91**, 187–198 (2017)
- [3] Xie, K., Zhang, X., Sui, S., Ye, F. & Dai J. Exploring and applying the substrate promiscuity of a C-glycosyltransferase in the chemo-enzymatic synthesis of bioactive C-glycosides. *Nat. Commun.* **11**, 5162 (2020)
- [4] Liu, X. et al. Characterization of a caffeoyl-CoA O-methyltransferase like enzyme involved in biosynthesis of polymethoxylated flavones in *Citrus reticulata*. *J. Exp. Bot.* 2020, **71**, 3066–3079 (2020)
- [5] Functional characterization of two new members of the caffeoyl CoA O-methyltransferase-like gene family from *Vanilla planifolia* reveals a new class of plastid-localized O-methyltransferases. *Plant Mol. Biol.* **76**, 475–488 (2011)

To Reviewer #2:

Reviewer comments:

The authors have carefully revised the manuscript. My queries have been fully addressed. The revised manuscript has remarkably improved quality. Thus, I suggest it be published in its current form.

Response #2:

Thanks for the helpful advices and positive comments.

To Reviewer #3:

Reviewer comments:

All of my concerns have been answered.

Response #3:

Thanks for the previous comments and helpful suggestions.

Reviewers' Comments:

Reviewer #1:

Remarks to the Author:

In this revised version, the authors have appropriately revised the manuscript, and my concerns have been fully addressed.

I noticed two minor errors in the added Supplementary Table 8.

1) The reference 2 (cited for supplementary information) is not for OsCGT but for maize CGT.

Reference for OsCGT would be Brazier-Hicks, M. et al. (2009).

2) "FeCGT" would be "FcCGT".

Please check and correct them.

To Reviewer #1:

Comments:

In this revised version, the authors have appropriately revised the manuscript, and my concerns have been fully addressed.

I noticed two minor errors in the added Supplementary Table 8.

1) The reference 2 (cited for supplementary information) is not for OsCGT but for maize CGT. Reference for OsCGT would be Brazier-Hicks, M. et al. (2009).

2) "FeCGT" would be "FcCGT".

Please check and correct them.

Response #1:

We are appreciated for this reviewer for his/her kind reminding and previous helpful suggestions.

For the two minor errors in Supplementary Table 8, we have corrected the supplementary reference 2 to the right reference (Brazier-Hicks, M. et al., 2009), and the typo "FeCGT" is also revised to "FcCGT" according to the suggestion.

Finally, we would like to again thank all the reviewers for their excellent review and constructive suggestions. Many thanks!